# Functional antagonism between ΔNp63α and GCM1 regulates human trophoblast stemness and differentiation

Liang-Jie Wang[1], Chie-Pein Chen[2], Yun-Shien Lee[3], Pui-Sze Ng[1], Geen-Dong Chang[4], Yu-Hsuan Pao[1], Hsiao-Fan Lo[1], Chao-Hsiang Peng[1], Mei-Leng Cheong[5] & Hungwen Chen [1,4✉]

The combination of EGF, CHIR99021, A83-01, SB431542, VPA, and Y27632 (EGF/CASVY) facilitates the derivation of trophoblast stem (TS) cells from human blastocysts and first-trimester, but not term, cytotrophoblasts. The mechanism underlying this chemical induction of TS cells remains elusive. Here we demonstrate that the induction efficiency of cytotrophoblast is determined by functional antagonism of the placental transcription factor GCM1 and the stemness regulator ΔNp63α. ΔNp63α reduces GCM1 transcriptional activity, whereas GCM1 inhibits ΔNp63α oligomerization and autoregulation. EGF/CASVY cocktail activates ΔNp63α, thereby partially inhibiting GCM1 activity and reverting term cytotrophoblasts into stem cells. By applying hypoxia condition, we can further reduce GCM1 activity and successfully induce term cytotrophoblasts into TS cells. Consequently, we identify mitochondrial creatine kinase 1 (CKMT1) as a key GCM1 target crucial for syncytiotrophoblast differentiation and reveal decreased CKMT1 expression in preeclampsia. Our study delineates the molecular underpinnings of trophoblast stemness and differentiation and an efficient method to establish TS cells from term placentas.

[1] Institute of Biological Chemistry, Academia Sinica, Nankang, Taipei 115, Taiwan. [2] Division of High Risk Pregnancy, Mackay Memorial Hospital, Taipei 104, Taiwan. [3] Department of Biotechnology, Ming Chuan University, Taoyuan 333, Taiwan. [4] Graduate Institute of Biochemical Sciences, National Taiwan University, Taipei 106, Taiwan. [5] Department of Obstetrics and Gynecology, Cathay General Hospital, Taipei 106, Taiwan. ✉email: hwchen@gate.sinica.edu.tw

Trophectoderm (TE) is the earliest differentiated cell lineage from a fertilized egg. TE contains a population of trophoblast stem (TS) cells on the outer surface of blastocyst. After implantation, TS cells proliferate and differentiate into different trophoblast subtypes to form the placenta. Human placenta is composed of villous tissues immersed in the maternal blood and attaching the placenta onto the uterus. The epithelial compartment of placental villi contains TS cell-like mononuclear cytotrophoblasts (CTBs), which may differentiate into a multi-nucleated syncytiotrophoblast (STB) layer or highly motile extravillous trophoblasts (EVTs). The STB layer mediates the exchange of nutrient, gas, and waste between fetus and mother and produces important hormones and growth factors. EVTs invade and migrate into the uterine decidua where they protect the fetus against maternal immune surveillance and remodel uterine spiral arteries to establish uteroplacental circulation. During placental development, STBs continuously undergo apoptosis and shed off into the circulation and are replenished by new STBs differentiated from CTBs. The maintenance of CTB stemness and differentiation into STB is of critical importance for pregnancy and the mechanism controlling this transition remains to be fully elucidated.

Studies on mouse placental development have identified key factors and signaling pathways that control TS cell maintenance, self-renewal, and differentiation. TEAD4 transcription factor and HIPPO signaling are required for TE specification, whereas the Cdx2, Eomes, and Elf5 transcription factors and Fgf and Tgf-β signaling are crucial for the maintenance of the TS cell niche[1–4]. Specifically, nuclear localization of TEAD4 in outer, but not inner, blastomeres regulates the commitment of TE and promotes the self-renewal of TS cells[5,6]. The TEAD4 cofactor YAP maintains trophoblast stemness by transactivation of cell cycle regulators and stemness-associated genes[7]. In addition to blastocysts, mouse TS cells can be generated from extra-embryonic ectoderm of post-implantation embryos in the presence of FGF4 and fetal fibroblast-conditioned medium[8]. Models of human TS cells have been established from human embryonic stem (hES) cells in BMP4 and feeder-conditioned medium that favor the differentiation of trophoblast lineages[9,10]. These hES-derived trophoblast-like cells express CDX2, hCGβ, and HLA-G, but exhibit low proliferative activity and fail to express ELF5 and *C19MC*, which is a miRNA cluster highly expressed in human trophoblasts[11]. In this regard, the JEG3 choriocarcinoma cell line expressing CDX2, ELF5, and *C19MC* exhibits some characteristics of TS cells[11].

Recent advances have identified small molecules and growth factors that create cell culture conditions and mimic tissue niche environments for epithelial stem cells. For example, the proliferative capacity of epithelial stem cells is significantly increased in the presence of 3T3 feeder cells plus the Rho-associated protein kinase (ROCK) inhibitor Y27632[12]. Dual inhibition of SMAD signaling by blockade of the TGF-β pathway with A83-01 and the BMP pathway with DMH-1 enhances stable propagation of human and mouse epithelial basal cell populations[13]. Combined treatment with glycogen synthase kinase 3 inhibitor CHIR99021 and histone deacetylase (HDAC) inhibitor valproic acid (VPA) promotes self-renewal of Lgr5+ small intestinal stem cells[14]. As trophoblasts are of epithelial origin, leveraging this knowledge, human TS cells have recently been derived from first-trimester CTBs (TS$^{CT}$ cells) or blastocysts (TS$^{blast}$ cells) using EGF, CHIR99021, A83-01, SB431542, VPA, and Y27632 (hereafter named EGF/CASVY in this study)[15]. EGF/CASVY cocktail can also facilitate the derivation of TS cells from naïve hES cells (TS$^{naive}$ cells) or naïve induced pluripotent stem cells (iTS cells)[15–19]. TS$^{naive}$ and iTS cells exhibit cellular and molecular phenotypes similar to that of TS$^{CT}$ and TS$^{blast}$ cells. How EGF/CASVY cocktail elicits the induction of a TS cell-like state is not known. For unknown reason, EGF/CASVY fails to establish TS cells from term CTBs[15]. The term-CTB resistance to chemical reprogramming poses a technical and ethical barrier to develop TS cells as an abundant source of stem cells for autologous cell therapies. A better understanding of TS cell maintenance and differentiation as well as the molecular targets of EGF/CASVY may facilitate the derivation of term TS cells for research and clinical use.

Glial cells missing 1 (GCM1) is a master transcriptional regulator of trophoblast differentiation. GCM1 activates the expression of *ERVW-1* (also known as *Syncytin-1*), *ERVFRD-1*, *HTRA4*, and *CGB* (also known as *hCGβ*) genes to promote trophoblast fusion, invasion, and hormone production critical for STB and EVT differentiation[20–23]. GCM1-dependent human trophoblast differentiation is regulated by the cAMP signaling pathway. In this context, cAMP stimulates the PKA-CBP/DUSP23 and Epac1-CaMKI-SENP1/HDAC5 signaling cascades to phosphorylate and activate GCM1[24–27]. On the other hand, hypoxia destabilizes GCM1 and triggers its degradation[28]. Inadequate GCM1 expression is linked to preeclampsia (PE), a clinical condition caused by poor placentation[29,30]. In contrast to GCM1-mediated trophoblast differentiation, much less is known about the mechanism that maintains trophoblast stemness. p63 is a family member of the p53 tumor suppressor. ΔNp63α, an isoform of p63 transcription factor produced by alternative promoter usage and splicing, is highly expressed in TS cell-like CTBs[31,32]. Importantly, ΔNp63α levels in CTBs are decreased during STB differentiation and ΔNp63α overexpression inhibits hCG expression in differentiating CTBs and cell migration of JEG3 cells, implying a role for ΔNp63α in regulation of STB and EVT differentiation[32,33]. ΔNp63α is also predominantly expressed in the stem cells of stratified epithelia[34]. These observations suggested a possible role of ΔNp63α in TS cell maintenance.

Here we report that GCM1 and ΔNp63α have antagonistic activities in TS cell maintenance and differentiation and their relative expression determines the induction efficiency of TS cell-like CTBs. We further show that EGF/CASVY suppresses GCM1 expression by upregulation of ΔNp63α activity, promoting trophoblast stemness in term CTBs. By further reducing GCM1 levels by hypoxia, we successfully convert term CTBs into TS cells, denoted as TS$^{Term}$ cells. RNA-sequencing analysis of wild-type and *GCM1*-knockout TS$^{Term}$ cells reveals *CKMT1A* and *-1B* as GCM1 target genes crucial for STB differentiation. Supporting its physiological and clinical relevance, CKMT1 expression is decreased in preeclamptic placentas. Our results suggest that trophoblast stemness and differentiation is controlled by the GCM1-ΔNp63α antagonism, whereby a combination of hypoxia and EGF/CASVY can induce a ΔNp63α-positive and GCM1-negative cell state that enables the generation of TS$^{Term}$ cells. The efficient production of TS$^{Term}$ cells may open new avenues for a better understanding of trophoblast differentiation and the pathogenesis of pregnancy-associated disorders as well as an abundant source of stem cells for clinical use.

## Results

**Expression of GCM1 and ΔNp63α in placenta.** We studied the expression of GCM1 and ΔNp63α in human placentas by immunohistochemistry (IHC). GCM1 expression was detected in the differentiated EVTs and STBs of GA7, GA20, and term placentas, whereas ΔNp63α was mainly expressed in the stem cell-like CTBs of the three gestational ages tested (Supplementary Fig. 1a). Very few GA20 and term CTBs expressing ΔNp63α were found, suggesting a loss of ΔNp63α in more advanced stages of pregnancy (Supplementary Fig. 1a). Of note, careful examination revealed ΔNp63α-positive nuclei in the STB layer of GA20 and

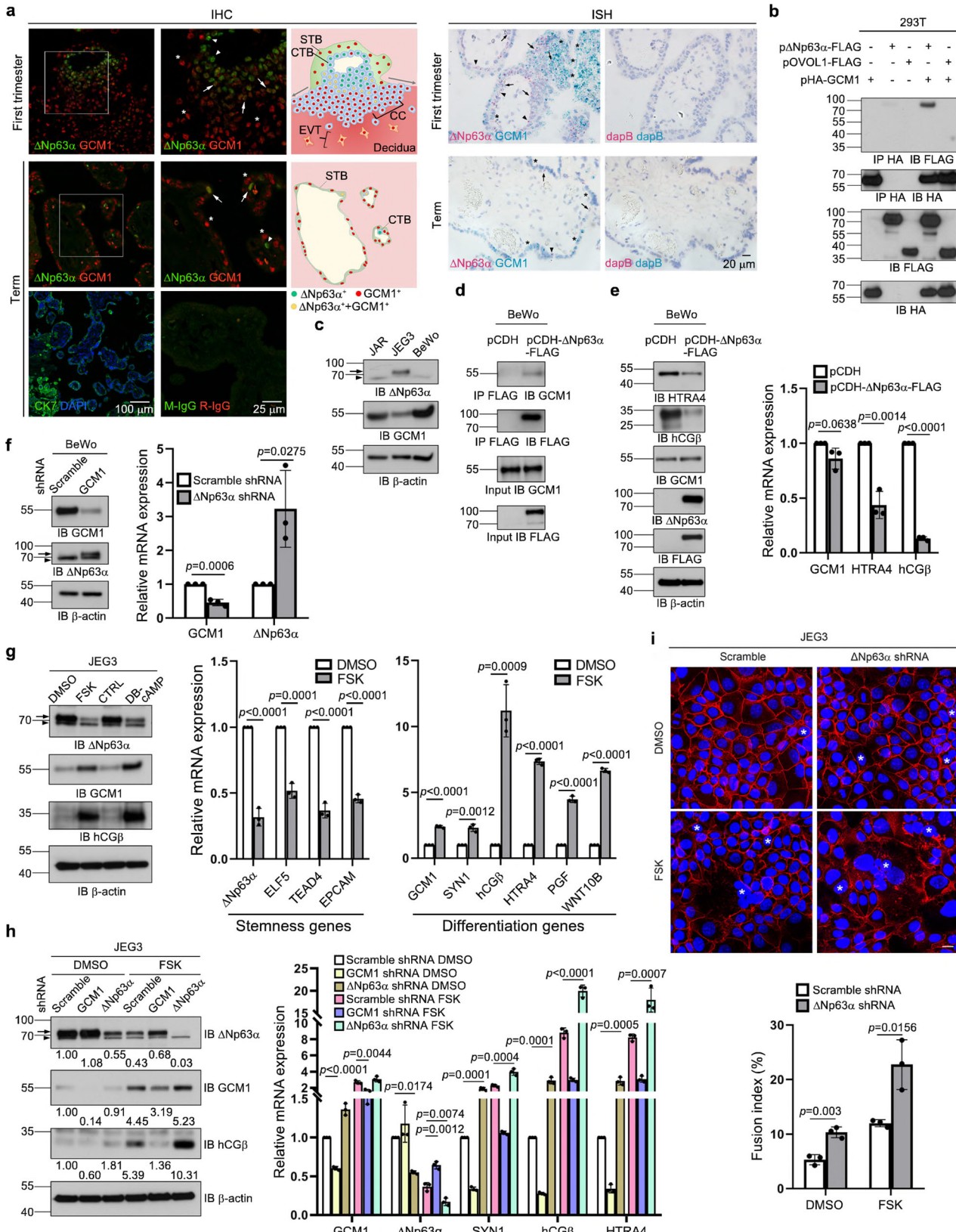

term placentas (Supplementary Fig. 1a). The expression of ΔNp63α in the differentiated STB layer prompted us to compare the expression patterns of GCM1 and ΔNp63α by double immunofluorescence staining. We observed ΔNp63α-expressing CTBs and GCM1-expressing EVTs in the proximal and distal regions of the GA7 cell column, respectively (Fig. 1a, left).

Interestingly, ΔNp63α and GCM1 double-positive trophoblasts were detected in the transition zone between the ΔNp63α-expressing CTBs and the GCM1-expressing EVTs (Fig. 1a, left). Coexpression of ΔNp63α and GCM1 was also detected in the CTBs subjacent to or in the nuclei of term STBs (Fig. 1a, left), which was corroborated by chromogenic IHC double staining

**Fig. 1 Physical and functional interaction between ΔNp63α and GCM1. a** Expression of ΔNp63α and GCM1 in human placental trophoblasts. First-trimester and term placental sections were subjected to immunohistochemistry (IHC) and RNAscope in situ hybridization (ISH) for analysis of ΔNp63α and GCM1 expression. Boxed areas are shown at higher magnification at right. Arrowhead, asterisk, and arrow indicate trophoblasts expressing ΔNp63α, GCM1, and both factors, respectively. Schematic illustrations of the IHC images are presented. CC, cell column. CTB, cytotrophoblast. STB, syncytiotrophoblast. EVT, extravillous trophoblast. Placental sections were stained with CK7 Ab for trophoblasts and normal mouse IgG (M-IgG) and rabbit IgG (R-IgG) for negative controls in IHC or hybridized with dapB probes for negative controls in ISH. **b** Interaction between GCM1 and ΔNp63α, but not OVOL1. **c** Expression of GCM1 and ΔNp63α in human JAR, JEG3, and BeWo trophoblast cell lines. **d** Interaction of ΔNp63α-FLAG and endogenous GCM1 in BeWo cells stably expressing ΔNp63α-FLAG. BeWo cells were transduced with lentiviruses harboring pCDH-ΔNp63α-FLAG. After puromycin selection, the ΔNp63α-FLAG-expressing BeWo cells were subjected to coimmunoprecipitation analysis using FLAG mAb and GCM1 Ab. **e** Suppression of GCM1 target gene expression in ΔNp63α-FLAG-expressing BeWo cells. **f**, Enhancement of ΔNp63α expression in BeWo cells by *GCM1* knockdown. **g** Suppression of trophoblast stemness gene expression in the differentiation of JEG3 cells stimulated by FSK or DB-cAMP. CTRL, control buffer. **h** Enhancement of trophoblast differentiation gene expression in FSK-stimulated JEG3 cells by *ΔNp63α* knockdown. The numbers underneath indicate the protein band intensities normalized against β-actin. Arrow and arrowhead in (**c**) and (**f–h**) denote the ΔNp63α band and a non-specific band, respectively. **i**, Enhancement of FSK-stimulated fusion of JEG3 cells by *ΔNp63α* knockdown. Scramble control and *ΔNp63α*-knockdown JEG3 cells treated with or without FSK were subjected to immunofluorescence microscopy with E-cadherin Ab. Syncytia are marked with asterisks. Scale bar, 20 μm. Cell fusion was quantified by fusion index. Data are presented as mean ± SD of independent experiments ($n = 3$ for **e–i**). Differences were assessed by unpaired two-tailed Student's *t*-test (**e–g**, **i**) or one-way ANOVA with Tukey's post hoc test (**h**). Source data are provided in the Source Data file.

(Supplementary Fig. 1b). In addition, coexpression of ΔNp63α and GCM1 transcripts in first-trimester and term CTBs was confirmed by RNAscope in situ hybridization (ISH, Fig. 1a, right). Proximal cell column CTBs proliferate and differentiate into EVTs that migrate into the distal cell column and decidua; villous CTBs undergo cell fusion and differentiate into STBs. The ΔNp63α- and GCM1-coexpressing trophoblasts may be in a transition state from stem cell-like CTBs to differentiated EVTs or STBs (Fig. 1a, left).

**Physical and functional interaction between GCM1 and ΔNp63α.** We next investigated potential interaction between GCM1 and ΔNp63α using transient expression in 293 T cells. By coimmunoprecipitation analysis, specific interaction was detected between GCM1 and ΔNp63α, but not OVOL1, which is a zinc finger-containing transcription factor regulating trophoblast fusion[35] (Fig. 1b). The GCM1-interacting domain in ΔNp63α was mapped to the region between amino acids 274 and 447 in the ΔNp63α polypeptide, which corresponds to the oligomerization domain (OD) (Supplementary Fig. 1c, left). Likewise, the ΔNp63α-interacting domain in GCM1 was mapped to the region between amino acids 167 and 349 in the GCM1 polypeptide, which harbors the transactivation domain 1 (Supplementary Fig. 1c, middle). By GST pull-down assays, physical interaction was detected between recombinant GCM1-FLAG and GST-OD (Supplementary Fig. 1c, right), suggesting that GCM1 likely directly interacts with ΔNp63α.

The GCM1 and ΔNp63α protein levels were surveyed in three different model human trophoblast cell lines. While JAR and BeWo cells express higher levels of GCM1 and barely express ΔNp63α, JEG3 cells express higher levels of ΔNp63α and low abundance of GCM1 (Fig. 1c), suggesting an inverse relationship between ΔNp63α and GCM1 expression in trophoblasts. Using lentiviral transduction followed by puromycin selection or GFP sorting, we found stable expression of ΔNp63α-FLAG in BeWo cells revealed an interaction with endogenous GCM1 (Fig. 1d) and led to decreased expression of GCM1 target genes, *HTRA4* and *hCGβ* (Fig. 1e and Supplementary Fig. 1d). The latter observation could be restored by GCM1 overexpression (Supplementary Fig. 1e). Conversely, knocking down *GCM1* in BeWo cells resulted in elevated ΔNp63α expression (Fig. 1f). JEG3 cells uniquely expressing both GCM1 and ΔNp63α, reminiscent of the GCM1 and ΔNp63α double-positive trophoblasts in GA7 cell column or term STBs (Fig. 1a). We treated JEG3 cells with the cAMP stimulant forskolin (FSK) or dibutyryl-cAMP (DB-cAMP) to activate GCM1 and studied the effect on the expression of

ΔNp63α and genes associated with trophoblast stemness and STB differentiation. Expression of *GCM1* and its differentiation-related target genes was increased as expected, whereas expression of *ΔNp63α* and trophoblast stemness genes *ELF5*, *TEAD4*, and *EPCAM* was decreased (Fig. 1g). The effect of FSK on ΔNp63α expression was compromised by *GCM1* knockdown in JEG3 cells confirming the role of GCM1 in the repression of ΔNp63α expression during STB differentiation (Fig. 1h). Furthermore, *ΔNp63α* knockdown led to decreased *ELF5*, *TEAD4*, and *EPCAM* expression and increased *GCM1* and *hCGβ* expression in JEG3 cells (Supplementary Fig. 1f), suggesting a critical role of ΔNp63α in maintaining trophoblast stemness gene expression and inhibiting differentiation. The role of ΔNp63α in suppression of trophoblast fusion, a critical feature of trophoblast differentiation, was assessed by E-cadherin immunostaining on cell-cell boundaries. Indeed, FSK stimulation-induced JEG3 cell fusion was further enhanced by *ΔNp63α* knockdown (Fig. 1i), supporting that ΔNp63α inhibits GCM1 activity and thereby GCM1-mdeiated trophoblast differentiation. These results suggested that the reciprocal and antagonistic regulation of GCM1 and ΔNp63α activities is important for trophoblast differentiation.

**ΔNp63α downregulates GCM1 activity and vice versa.** To investigate the mechanism underlying the antagonistic activity of GCM1 and ΔNp63α, we studied whether ΔNp63α directly inhibits the DNA-binding and/or transcriptional activity of GCM1 by mammalian two-hybrid assays in 293 T cells and found that neither activity is affected by ΔNp63α (Supplementary Fig. 2). Therefore, it is unlikely that ΔNp63α directly inhibits GCM1 activity.

GATA3 is a ΔNp63α downstream target that interacts with GCM1 and inhibit its transcriptional activity[34,36,37]. Indeed, we demonstrated that *GATA3* promoter was activated by ΔNp63α in Hep3B cells (Supplementary Fig. 3a and b) and GATA3 expression was elevated in the BeWo cells stably expressing ΔNp63α-FLAG or decreased in the *ΔNp63α*-knockdown JEG3 cells (Fig. 2a). GATA3 was shown to suppresses GCM1-dependent *HTRA4* promoter activity[36]. We investigated the effect of *ΔNp63α* knockdown on the luciferase activity directed by pHTRA4-1Kb in JEG3 cells. The *HTRA4* promoter activity was increased by *ΔNp63α* knockdown – an effect reversed by GATA3-FLAG overexpression (Fig. 2b). These results suggested that ΔNp63α indirectly inhibits GCM1 activity through the induction of GATA3.

We next determined how GCM1 inhibits ΔNp63α expression (Fig. 1f). Oligomerization is essential for the biological functions

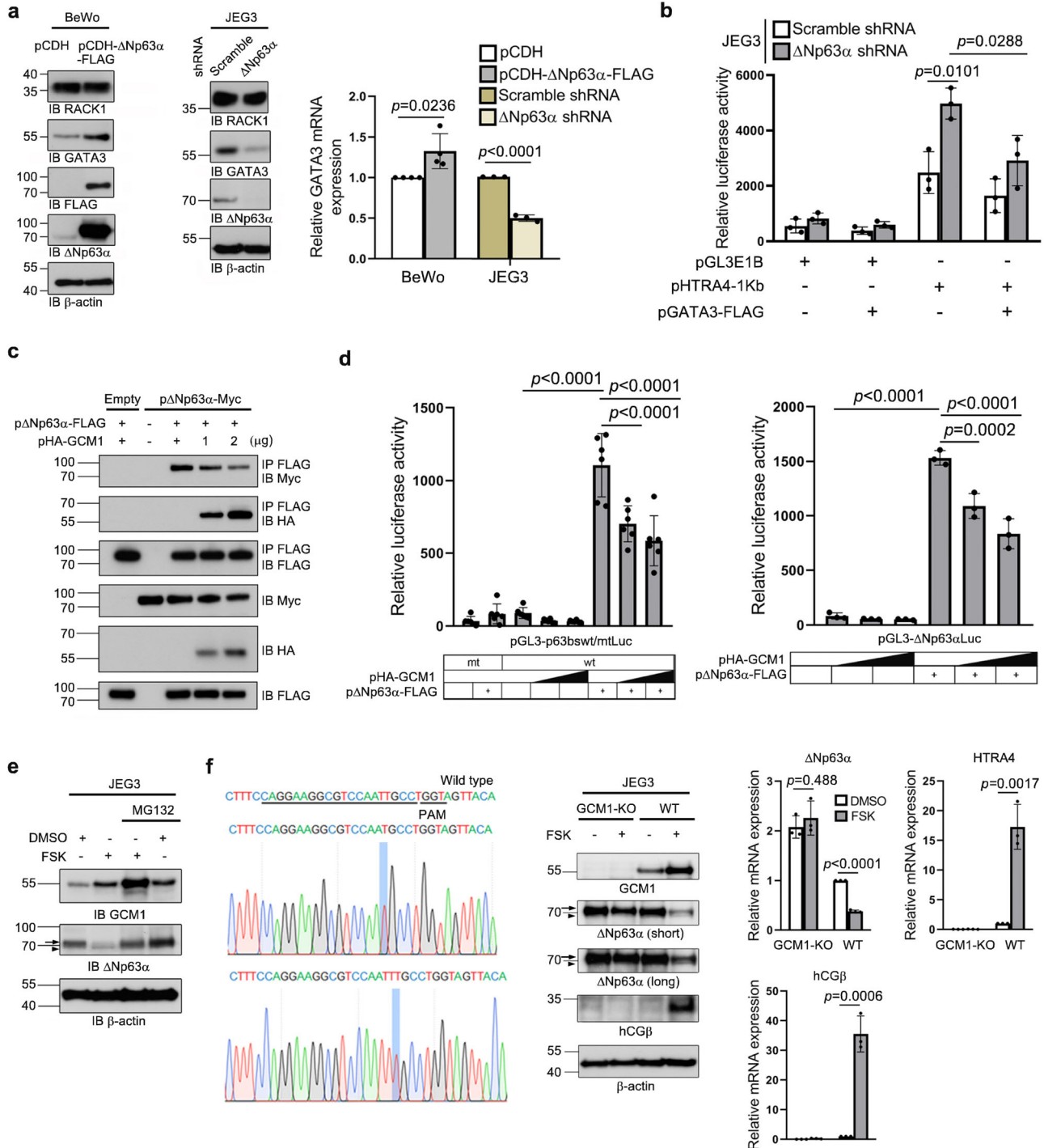

**Fig. 2 Reciprocal negative regulation between GCM1 and ΔNp63α. a** Activation of *GATA3* gene expression by ΔNp63α. The protein and transcript levels of GATA3 was measured in ΔNp63α-FLAG-expressing BeWo cells and *ΔNp63α*-knockdown JEG3 cells by immunoblotting and quantitative RT-PCR analyses. As an irrelevant control, expression of the RACK1 scaffold protein was not affected by ΔNp63α. **b** Suppression of *HTRA4* promoter activity by ΔNp63α through GATA3. Scramble control or *ΔNp63α*-knockdown JEG3 cells were transfected with pHTRA4-1Kb alone or plus pGATA3-FLAG. At 48 h post-transfection, cells were harvested for luciferase assays. **c** Impairment of ΔNp63α oligomerization by GCM1. **d** Suppression of ΔNp63α-mediated transcriptional activation by GCM1. Hep3B cells were transfected with pΔNp63α-FLAG and increasing amounts of pHA-GCM1 plus the reporter plasmid pGL3-p63bswtLuc or pGL3-p63bsmtLuc (left) or pGL3-ΔNp63αLuc (right). At 48 h post-transfection, cells were harvested for luciferase assays. **e** Stimulation of proteasomal degradation of ΔNp63α by FSK in JEG3 cells. **f** Inhibition of ΔNp63α activity by FSK is abrogated in *GCM1*-KO JEG3 cells generated by CRISPR/Cas9. Sequence of sgRNA is underlined and the mutant sequences in *GCM1* locus are highlighted. Arrow and arrowhead in (**e**) and (**f**) denote the ΔNp63α band and a non-specific band, respectively. Data are presented as mean ± SD of independent experiments ($n = 4$ for BeWo and $n = 3$ for JEG3 in (**a**); $n = 3$ in (**b**), (**f**); $n = 6$ (left) and $n = 3$ (right) in **d**). Differences were assessed by unpaired two-tailed Student's *t*-test (**a**, **b**, **f**) or two-way ANOVA with Tukey's post hoc test (**d**). Source data are provided in the Source Data file.

of p53 family members in regulation of cell cycle and development[38]. In transient expression experiments, we demonstrated that GCM1 blocks ΔNp63α oligomerization as the interaction between ΔNp63α-FLAG and ΔNp63α-Myc is interrupted by increasing amounts of HA-GCM1 in 293 T cells (Fig. 2c). Along this line, the p63 reporter construct pGL3-p63bswtLuc, which harbors multiple p63-binding sites, was activated by ΔNp63α-FLAG, but inhibited by HA-GCM1 in a dose-dependent manner in Hep3B cells (Fig. 2d, left). ΔNp63α is subject to autoregulation[39,40] and we found that ΔNp63α-FLAG indeed activates the ΔNp63α promoter reporter construct pGL3-ΔNp63αLuc. Importantly, ΔNp63α promoter activity was also inhibited by HA-GCM1 in a dose-dependent manner in Hep3B cells (Fig. 2d, right). Therefore, the interaction between GCM1 and ΔNp63α results in suppression of ΔNp63α transcriptional activity and its autoregulation.

In addition, we demonstrated that FSK inhibits ΔNp63α activity through proteasomal degradation because the ΔNp63α protein level in the FSK-treated JEG3 cells was restored by MG132 (Fig. 2e). However, in vivo ubiquitination assay performed in JEG3 cells expressing HA-tagged ubiquitin showed that ubiquitination of ΔNp63α is not affected by GCM1 or FSK (Supplementary Fig. 4). These observations suggested that ubiquitin-independent proteasomal degradation may be involved in the enhancement of ΔNp63α degradation by FSK. We generated GCM1-knockout (KO) JEG3 cells by the CRISPR/Cas9 system and then demonstrated that FSK-mediated upregulation of hCGβ and HTRA4 and downregulation of ΔNp63α were blunted in the GCM1-KO JEG3 cells (Fig. 2f). Collectively, these results indicated that GCM1 inhibits ΔNp63α activity by blockade of ΔNp63α oligomerization and autoregulation and by promoting ΔNp63α degradation.

**EGF/CASVY targets GCM1 and ΔNp63α expression.** EGF/CASVY was used to generate TS^CT and TS^blast cells[15]; however, the underlying mechanism is not known. In BeWo cells, which express GCM1 but little ΔNp63α, we observed that EGF/CASVY treatment increases the expression of ΔNp63α and trophoblast stemness genes ELF5, TEAD4, and EPCAM, and concomitantly decreases the expression of GCM1 and trophoblast differentiation genes hCGβ, HTRA4, PGF, and WNT10B (Fig. 3a). Induction of the trophoblast stemness genes by EGF/CASVY was very likely associated with ΔNp63α because ΔNp63α knockdown in BeWo cells results in a significant decrease in ELF5, TEAD4, and EPCAM expression (Supplementary Fig. 5). These findings indicated that EGF/CASVY targets the GCM1-ΔNp63α antagonistic signaling axis to induce BeWo cells into a TS cell-like state.

To further test this hypothesis, we determined whether EGF/CASVY can regulate GCM1 and ΔNp63α expression in the CTBs sorted from term placentas and revert them into TS cells. Immunostaining showed that the initial purified ITGA6-positive term CTBs were composed of cells expressing ΔNp63α and various amounts of GCM1 (Fig. 3b, upper). After treatment with EGF/CASVY for 5 days, the population became dominated by ΔNp63α-expressing cells with only a few cells weakly expressing GCM1 (Fig. 3b, lower). 14-day treatment further elevated ΔNp63α expression while GCM1 was barely detectable (Fig. 3c, upper). These findings suggested that EGF/CASVY induces trophoblast stem cell factor ΔNp63α but represses differentiation master GCM1. Importantly, upon removal of EGF/CASVY for 7 days, term CTBs underwent differentiation and formed syncytia coexpressing GCM1 and hCGβ (Fig. 3c, lower). Therefore, EGF/CASVY treatment induces term CTBs to acquire some properties of TS cells.

However, attempts to establish TS cells from term placentas by extended culture of ITGA6-positive term CTBs with EGF/CASVY

failed to reach the third passage (P3) – a result similar to Okae's findings[15]. We speculated that EGF/CASVY is not sufficient to completely repress GCM1 activity for maintaining trophoblast stemness because residual GCM1 expression remains detectable in the ITGA6-positive term CTBs treated with EGF/CASVY for 14 days (Fig. 3c, upper). We have previously reported that hypoxia suppresses GCM1 at the transcriptional and post-translational levels[28]. We investigated the effect of hypoxia on BeWo cells and P0 ITGA6-positive term CTBs. As shown in Fig. 3d, expression of GCM1 and its target genes was decreased whereas expression of ΔNp63α and/or MKI67 was increased in BeWo cells and P0 ITGA6-positive term CTBs by hypoxia. Culture of ITGA6-positive term CTBs with EGF/CASVY under hypoxia maintained the cells in a proliferative state beyond P3.

**Derivation of TS cells from term placentas.** In fact, the combination of hypoxia and EGF/CASVY enabled us to maintain the ITGA6-positive term CTBs from different placentas for over 20 passages. We named these cells as TS^Term cells, which expressed key markers of TS cells including ΔNp63α, EPCAM, GATA3, TFAP2C, and MKI67 (Fig. 4a, b). Two TS^Term cell lines (#1 and #2) were subjected to chromosomal microarray and karyotype analyses and had normal chromosomal complements (Supplementary Fig. 6). Characterization of one TS^Term line (#2) demonstrated bipotential ability to differentiate into multi-nucleated and hCGβ-positive STBs (ST-TS^Term#2) or HLA-G-positive and migratory EVTs (EVT-TS^Term#2) in response to FSK or A83-01 and NRG1 under normoxia (Fig. 4c, d). The suppressive effects of hypoxia on GCM1 expression was confirmed by diminished GCM1 and hCGβ expression in ST-TS^Term#2 cells and reduced activation of the GCM1 promoter reporter construct E1bLUCGCM1-2K in TS^Term#1 cells under hypoxia (Fig. 4e, f). Interaction between ΔNp63α and GCM1 was studied in normoxic TS^Term#2 cells by in situ proximity ligation assays (PLAs). Compared with the negative control with normal rabbit IgG (R-IgG) and ΔNp63 mAb, a significant increase in the average PLA signal per nucleus was observed in the group with GCM1 Ab and ΔNp63 mAb, suggesting that ΔNp63α may interact with GCM1 in TS^Term#2 cells (Fig. 4g).

To study the functional interaction between GCM1, ΔNp63α, and GATA3 in TS^Term cells, we demonstrated that ΔNp63α very likely regulates GATA3 expression in TS^Term#2 cells because ΔNp63α is associated with the GATA3 promoter by ChIP analysis and ΔNp63α knockdown decreases GATA3 expression (Supplementary Fig. 3c, d). Along this line, the expression of ΔNp63α and GATA3 was decreased when TS^Term#2 cells were differentiated into ST-TS^Term#2 cells (Supplementary Fig. 3e). The reciprocal regulation of ΔNp63α and GCM1 activities was tested in TS^Term#2 cells stably expressing ΔNp63α-FLAG. As shown in Fig. 4h, FSK-stimulated expression of hCGβ and Syncytin-1 (SYN1) genes and cell fusion was significantly compromised in the ΔNp63α-FLAG-expressing ST-TS^Term#2 cells. Correspondingly, overexpression of exogenous GATA3-FLAG in TS^Term#2 cells also compromised the differentiation of ST-TS^Term#2 cells in terms of hCGβ expression (Supplementary Fig. 3f). Therefore, the ΔNp63α-GATA3 axis is downregulated and GCM1 is upregulated in TS^Term cells during STB differentiation, which can be counteracted by overexpression of ΔNp63α or GATA3. By subcutaneous injection of TS^Term cells into immunodeficient NOD-SCID mice for 10 days, CK7-positive cells were readily detected in lesions and some of them expressed hCGβ or HLA-G, suggesting that TS^Term cells are bipotential in vivo (Fig. 5a, b). These findings indicated that combination of hypoxia and EGF/CASVY cocktail can modulate the GCM1-ΔNp63α antagonism to fully reprogram term CTBs into TS cells.

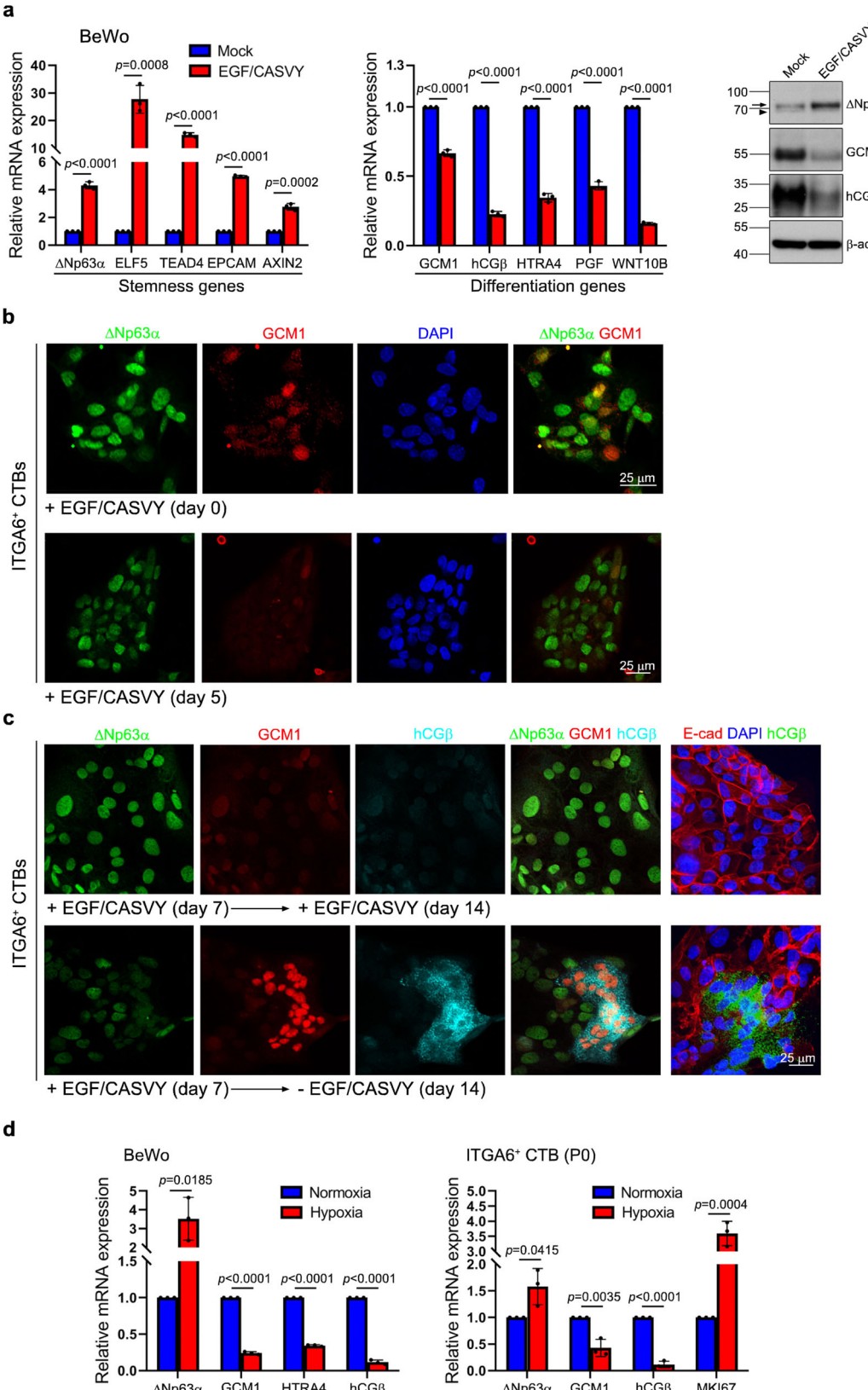

**Gene expression profiles of TS cells and their derivatives**. TS$^{Term}$ (#1-#3) cells and their derivative STBs (ST-TS$^{Term}$) and EVTs (EVT-TS$^{Term}$) as well as ITGA6-positive term CTBs (#1-#3) were subjected to RNA-sequencing (RNA-seq) analyses. We analyzed the correlation matrix for 18087 expressed genes in the RNA-seq datasets of this and Okae's studies using a PCA. The three-dimensional plot of the first three principal components by the matrix showed dissimilarities among different placental cell types, which could be categorized in four main clusters: (1) first-trimester and term CTBs, (2) TS$^{CT}$, TS$^{blast}$, and TS$^{Term}$ cells, (3) first-trimester STBs and the STBs derived from TS$^{CT}$, TS$^{blast}$, and TS$^{Term}$ cells, and (4) first-trimester EVTs and the EVTs derived

**Fig. 3 Regulation of trophoblast stemness by EGF/CASVY and hypoxia through the ΔNp63α-GCM1 antagonism. a** BeWo cells were mock-treated or treated with EGF/CASVY (i.e., cultured in incomplete or complete TS medium) for 24 h for analysis of trophoblast stemness or differentiation gene expression. *AXIN2*, a direct target of WNT signaling, was used as a control. Arrow and arrowhead denote the ΔNp63α band and a non-specific band, respectively. **b** ITGA6-positive term CTBs were cultured in complete TS medium for indicated periods of time and then subjected to immunofluorescence microscopy of GCM1 and ΔNp63α. **c** Expression of ΔNp63α, GCM1, and hCGβ in ITGA6-positive term CTBs in response to EGF/CASVY treatment and withdrawal. Note that E-cadherin staining reveals formation of hCGβ-expressing syncytium after EGF/CASVY withdrawal. **d**, Suppression of *GCM1* expression by hypoxia. BeWo cells and P0 ITGA6-positive term CTBs were cultured under normoxic or hypoxic conditions for three and seven days, respectively. Cells were then harvested for analysis of GCM1, ΔNp63α, HTRA4, hCGβ, and MKI67 transcripts. Data are presented as mean ± SD of independent experiments ($n = 3$ in **a**, **d**). Differences were assessed by unpaired two-tailed Student's *t*-test (**a**, **d**). Source data are provided in the Source Data file.

from TS$^{CT}$, TS$^{blast}$, and TS$^{Term}$ cells (Fig. 6a). Examination of the transcriptomic signatures identified 3296 genes that were differentially expressed (absolute fold change >3, $p < 0.05$) between ST-TS$^{Term}$ and TS$^{Term}$ cells and 2714 genes between EVT-TS$^{Term}$ and TS$^{Term}$ cells. Volcano plots of differentially expressed genes (DEGs) between differentiated trophoblasts and TS$^{Term}$ cells revealed significantly upregulated genes in different trophoblast lineages, e.g., *TEAD4*, *EPCAM*, *TP63*, *HAND1*, and *ITGA2* in TS$^{Term}$ cells, *LHB*, *ERVFRD-1*, *GCM1*, *CGA*, and *CGB5* in ST-TS$^{Term}$ cells, and *HLA-G*, *MMP2*, *ITGA1*, *GCM1*, and *FLT4* in EVT-TS$^{Term}$ cells (Fig. 6b). We merged the two groups of DEGs and a total of 4270 genes were identified as differentiation-related genes in STBs and EVTs. By Pearson correlation analysis, we found that TS$^{Term}$ cells are highly correlated with TS$^{CT}$ and TS$^{blast}$ cells (Fig. 6c). We identified 1015, 351, and 457 genes that were predominantly expressed in TS$^{Term}$, ST-TS$^{Term}$, and EVT-TS$^{Term}$ cells, respectively (fold change >3, $p < 0.05$). Most of these lineage-specific genes shared similar expression patterns with TS$^{CT}$ and TS$^{blast}$ cells and their derivative STBs and EVTs. For instance, *TP63*, *TEAD4*, and *ITGA2* were included in all the TS$^{Term}$, TS$^{CT}$, TS$^{blast}$ gene lists, *CGB5*, *LHB*, and *ERVFRD-1* in all the ST-TS gene lists, and *HLA-G*, *FLT4*, and *MMP2* in all the EVT-TS gene lists (Fig. 6d). Of the unmatched genes, we noted that many of them, such as *TMEM131L*, *ADGRL2*, *RFLNA*, *PRXL2A*, and *LARGE2* in the TS$^{Term}$ gene list, *ARLNC1*, *RIPOR2*, and *CGB3* in the ST-TS$^{Term}$ gene list, and *LHFPL6* in the EVT-TS$^{Term}$ gene list, are expressed in the CTB, STB, and EVT lineages derived from 3D-cultured human blastocysts by single-cell RNA-seq analysis[41] (Fig. 6d).

We compared the expression levels of some lineage marker genes in TS$^{Term}$, TS$^{CT}$, and TS$^{blast}$ cells and their derivative STBs and EVTs. The stemness-related genes examined were expressed in all three TS cell types in a comparable manner (Fig. 6e, left). Likewise, the examined STB- and EVT-related genes exhibited similar expression patterns and levels in the STBs and EVTs differentiated from all three TS cell types (Fig. 6, middle and right). Functional annotation of the lineage-specific gene lists indicated that pathways relating to WNT, cell cycle, and telomere maintenance may contribute to stem cell self-renewal and proliferation of TS$^{Term}$ cells. Genes related to glycoprotein hormone and peptide hormone biosynthesis and metabolism were overexpressed in the ST-TS$^{Term}$ cells. Extracellular matrix organization, integrin cell surface interaction, and epithelial to mesenchymal transition required for cell migration and invasion were upregulated in the EVT-TS$^{Term}$ cells (Fig. 6f). Thus, the gene expression patterns support TS$^{Term}$ cells as functional TS cells.

**Regulation of STB differentiation by creatine kinase.** To extend the physiological scope of STB derived from TS$^{Term}$ cells, we generated two *GCM1*-KO TS$^{Term}$ clones, TS2$^{GCM1-KO\#6}$ and TS2$^{GCM1-KO\#7}$. Both clones significantly lost bipotential differentiation capacities to STBs and EVTs in vitro (Supplementary

Fig. 7). The in vivo differentiation capacity of TS$^{T}$#2$^{GCM1-KO\#6}$ cells was also lost as tested in NOD-SCID mice (Fig. 5c). Furthermore, RNA-seq analysis of the STBs derived from TS$^{Term\#1}$, TS$^{Term\#2}$, and TS#1$^{GCM1-KO}$ cells revealed potential GCM1 target genes involved in STB differentiation. In combination with our previous ChIP-chip study of GCM1 target genes[21], we concentrated on mitochondrial creatine kinase 1A and -1B (*CKMT1A* and *-1B*), which catalyze the transfer of the γ-phosphate group of ATP to the guanidino group of creatine to produce phosphocreatine (Fig. 7a and Supplementary Fig. 8). We demonstrated that CKMT1 expression is increased in the STBs derived from TS$^{Term\#1}$, -#2, and -#3 cells (Fig. 7b) and that GCM1 upregulates *CKMT1* expression in STB differentiation because this upregulation is significantly reduced by *GCM1* knockout in TS$^{Term}$ and JEG3 cells (Fig. 7c). Notably, *CKMT1* expression was not significantly changed in the EVTs derived from TS$^{Term\#2}$ cells, suggesting that *CKMT1* is not involved in EVT differentiation (Fig. 7c). This notion was supported by IHC showing that CKMT1 is primarily expressed in the STB layer, but neither the CTBs subjacent to the STB layer nor the EVTs in the cell column (Fig. 7d). Subcellular localization demonstrated that CKMT1 expression is dramatically increased in the mitochondria of the STBs (ST-TS$^{Term\#2}$) differentiated from scramble control TS$^{Term\#2}$ cells compared with the undifferentiated TS$^{Term\#2}$ cells and the *CKMT1*-knockdown ST-TS$^{Term\#2}$ cells (Fig. 7e). Importantly, knocking down *CKMT1* impaired STB differentiation from TS$^{Term\#2}$ cells by inhibition of *hCGβ* and *LHB* expression and cell fusion (Fig. 7f, g). STB differentiation was also compromised in the TS$^{Term\#2}$ cells treated with the CKMT1 inhibitor cyclocreatine by decreasing the expression of *hCGβ*, *SYN1*, and *LHB* genes (Fig. 7h). Therefore, the creatine phosphate shuttle system is crucial for STB differentiation, which can be controlled through the GCM1-CKMT1 axis. Because defective trophoblast differentiation is associated with PE, we examined the microarray data of 77 control and 80 preeclamptic placentas in the GSE75010 public databases[42] for *CKMT1* expression. Interestingly, a significant decrease in the *CKMT1* gene expression was noted in the PE patients, which was corroborated by reduced CKMT1 expression in the preeclamptic STBs of different gestational ages by immunofluorescence microscopy (Fig. 7i).

## Discussion

Placental growth and development depend on the fine balance between self-renewal and differentiation of TS cells. The multinucleated STBs shed via apoptosis into the maternal circulation, which requires differentiation of the adjacent TS cell-like CTBs into STBs by cell fusion to maintain tissue homeostasis. An intriguing question is how TS cells are maintained and become terminally differentiated trophoblasts.

In the present study, we provided evidence that functional antagonism between ΔNp63α and GCM1 controls the cell fate of TS cell-like CTBs, where ΔNp63α maintains stemness while GCM1 promotes differentiation. IHC and ISH revealed trophoblasts

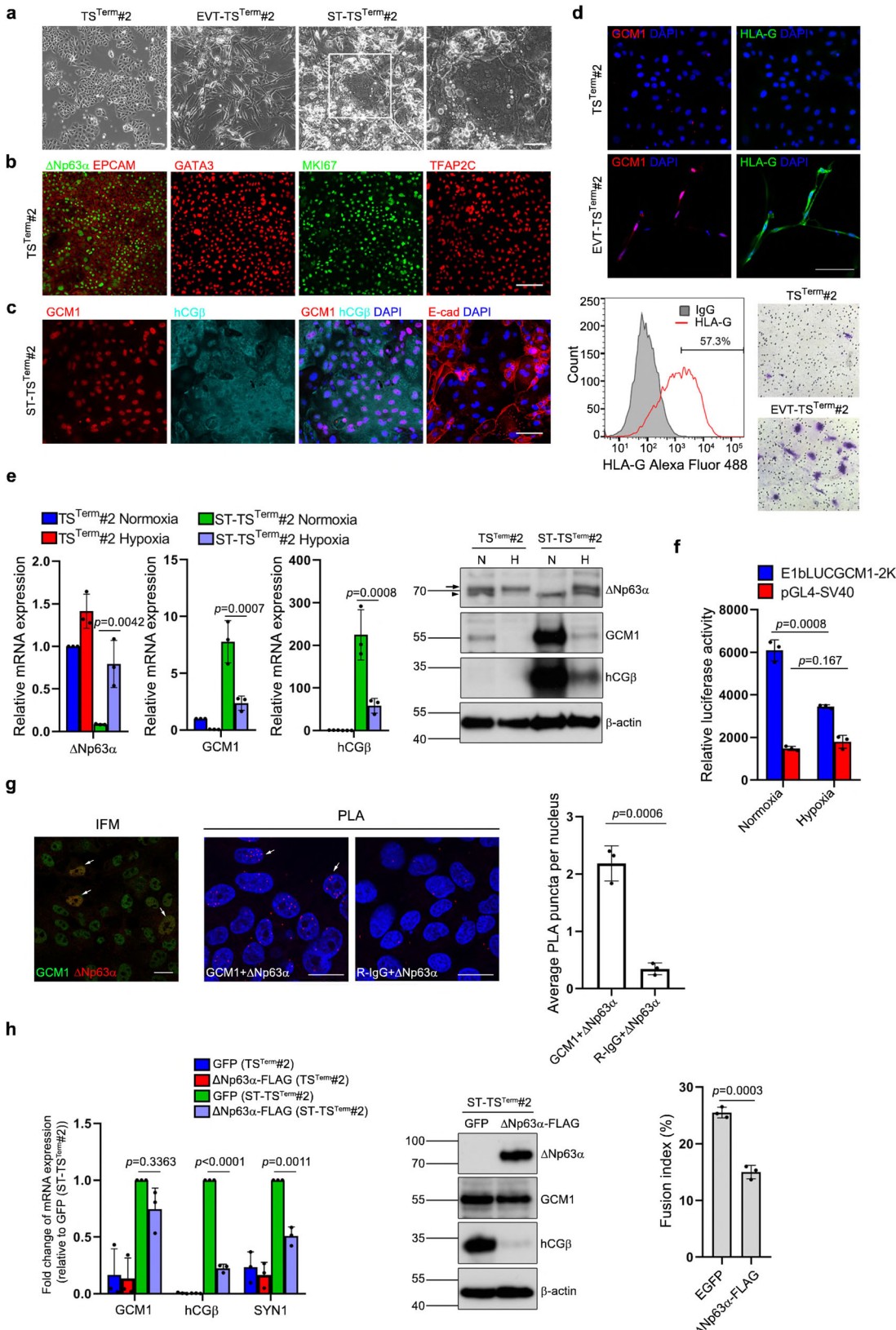

coexpressing ΔNp63α and GCM1 in the cell column of first-trimester placenta or in the STB layer of term placenta. We believed that these ΔNp63α- and GCM1-positive trophoblasts are most likely differentiating cells that transit from the ΔNp63α-positive TS cell-like CTBs into the differentiated GCM1-positive EVTs or STBs. In this scenario, we showed that ΔNp63α directly interacts with GCM1 and blocks trophoblast differentiation by diminishing GCM1 activity through GATA3. Upon stimulation by cAMP signaling, GCM1 is activated and induces the expression of *hCGβ*, *HTRA4*, and *SYN1* genes for STB differentiation, while inhibiting the expression of *ΔNp63α* and trophoblast stemness genes *ELF5*, *TEAD4*, and *EPCAM* (Fig. 1g). The suppression of ΔNp63α activity

**Fig. 4 Derivation of TS cells from term placentas. a** Images of TS$^{Term}$#2 cells and their derivative EVTs and STBs. TS$^{Term}$#2 cells derived from ITGA6-positive term CTBs were treated with FSK (ST medium) for five days or A83-01 and NRG1 (EVT medium) for 10 days for differentiation into STBs (ST-TS$^{Term}$#2) or EVTs (EVT-TS$^{Term}$#2), respectively. Magnification of the boxed syncytium in ST-TS$^{Term}$#2 cells is presented at right. **b** Expression of trophoblast stemness markers in TS$^{Term}$#2 cells. **c** Expression of STB markers and formation of syncytium in ST-TS$^{Term}$#2 cells. **d** Expression of EVT markers in EVT-TS$^{Term}$#2 cells. The HLA-G-positive EVT-TS$^{Term}$#2 cells were subjected to transwell invasion assays. Scale bar, 100 μm. **e** Regulation of GCM1 and ΔNp63α expression in TS$^{Term}$#2 cells and ST-TS$^{Term}$#2 by hypoxia. Arrow and arrowhead denote the ΔNp63α band and a non-specific band, respectively. **f** Suppression of *GCM1* autoregulation by hypoxia. TS$^{Term}$#1 cells were transfected with pGL4-SV40 or E1bLUCGCM1-2K and incubated under normoxia or hypoxia for 48 h. Cells were then harvested for luciferase assays. **g** Interaction of GCM1 and ΔNp63α in TS$^{Term}$ cells. TS$^{Term}$#2 cells were cultured under normoxia for 96 h and then subjected to immunofluorescence microscopy (IFM) or PLA using GCM1 Ab, ΔNp63 mAb, and normal rabbit IgG (R-IgG). Arrows point to cells coexpressing ΔNp63α and GCM1 in IFM or exhibiting PLA puncta. Average PLA signal per nucleus was calculated as total PLA puncta divided by the number of all nuclei in the field. Scale bar, 20 μm. **h** Suppression of STB differentiation by ΔNp63α. GFP-positive mock or ΔNp63α-expressing TS$^{Term}$#2 cells were treated with or without FSK for 72 h, followed by cell fusion assays and analysis of GCM1, hCGβ and SYN1 transcripts or proteins. Data are presented as mean ± SD of independent experiments ($n = 3$ in **e**–**h**). Differences were assessed by two-way ANOVA with Tukey's post hoc test (**e**, **h** (left)) or unpaired two-tailed Student's *t*-test (**f**, **g**, **h** (right)). Source data are provided in the Source Data file.

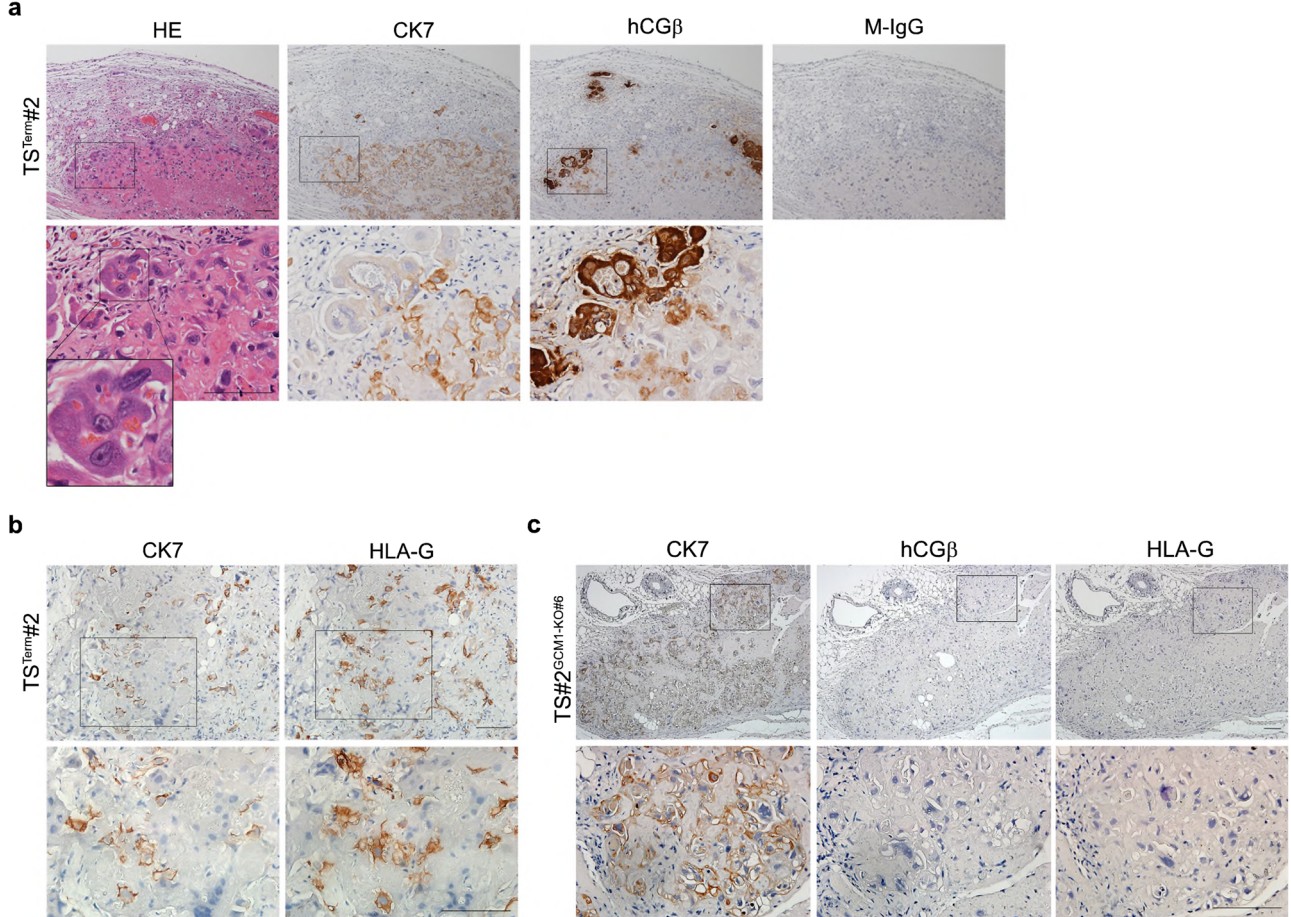

**Fig. 5 In vivo differentiation potential of WT and *GCM1*-KO TS$^{Term}$ cells.** NOD-SCID mice were subcutaneously injected with TS$^{Term}$#2 (**a**, **b**) or TS#2$^{GCM1-KO#6}$ (**c**) cells for 10 days. Sections of TS$^{Term}$#2 and TS#2$^{GCM1-KO#6}$ cell-derived lesions were subjected to hematoxylin and eosin (HE) staining and immunostaining of CK7, hCGβ, and HLA-G. Scale bar, 100 μm.

by GCM1 can be attributed to the binding of GCM1 to the OD of ΔNp63α, which impairs ΔNp63α oligomerization leading to ΔNp63α destabilization and reduction in *ΔNp63α* autoregulation. The reciprocal negative regulation between ΔNp63α and GCM1 prompted us to speculate that coexpression of both factors is a transient event during the differentiation of CTBs into STBs and EVTs in placenta. Because GCM1 promotes ΔNp63α degradation, the time window of ΔNp63α$^+$→GCM1$^+$/ΔNp63α$^+$→GCM1$^+$ transition is expected to be narrow. Indeed, the number of GCM1$^+$/ΔNp63α$^+$ CTBs is less than 10% of the ITGA6-positive term CTB population (Fig. 3b).

The mechanism underlying establishment and maintenance of TS$^{CT}$, TS$^{blast}$, TS$^{naive}$, and iTS cells by EGF/CASVY has not been clarified. The present study suggests that ΔNp63α and GCM1 are the molecular targets of EGF/CASVY. Specifically, EGF/CASVY induces dedifferentiation of BeWo cells by activation of *ΔNp63α* and trophoblast stemness gene expression, and thereby suppress GCM1 activity and differentiation-related target gene expression. Intriguingly, *ΔNp63α* knockdown impeded the induction of *ELF5*, *TEAD4*, and *EPCAM* by EGF/CASVY in BeWo cells (Supplementary Fig. 5) and downregulated the expression of *ELF5*, *TEAD4*, and *EPCAM* in JEG3 cells (Supplementary Fig. 1f). These

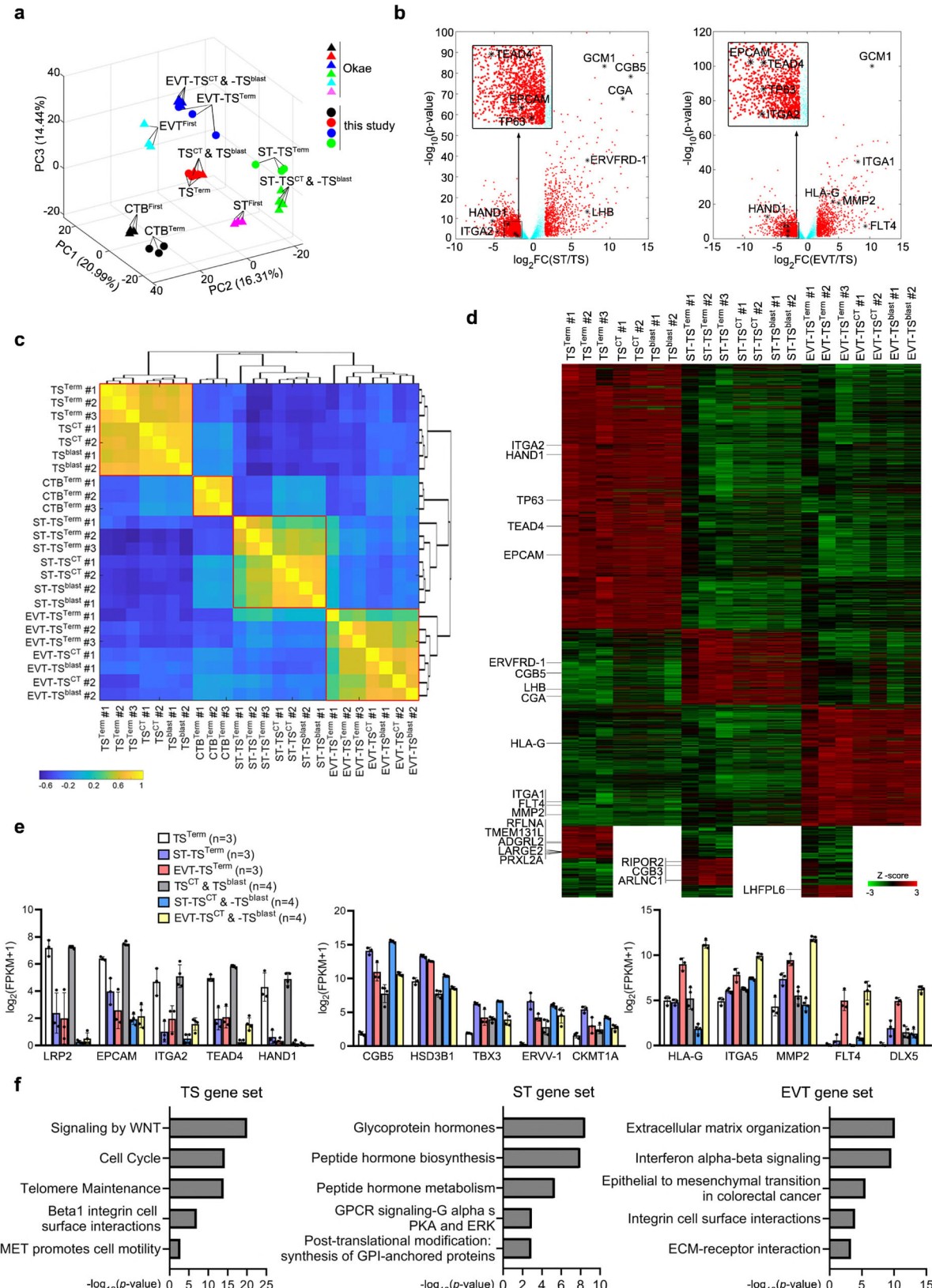

observations suggested a gene regulatory network of *ΔNp63α* and trophoblast stemness genes in the regulation of trophoblast stemness. EGF/CASVY also induces term CTBs into a population of ΔNp63α-positive CTBs. Because *ΔNp63α* overexpression and knockdown leads to decreased and elevated GCM1 activity, respectively (Figs. 1e, h, and 4h), it is highly possible that EGF/ CASVY stimulates ΔNp63α expression, which in turn suppresses GCM1 activity to redirect BeWo and term CTBs cells into a stem cell state. Nevertheless, residual GCM1 activity remains in term CTBs treated with EGF/CASVY, which could underlie the failure of chemical reprogramming of TS cells from term placentas[15]. We found that this hurdle could be readily overcome by hypoxia

**Fig. 6 Gene expression profiling of TS^Term cells and their derivative STBs and EVTs. a** PCA plot of the genes expressed in the different cell types. Each of the CTBs, TS cells, and TS cell-derived STBs and EVTs clusters together in first three dimensions. Data were collected from independent preparations of CTB^First ($n = 3$), EVT^First ($n = 3$), ST^First ($n = 3$), TS^CT & TS^blast ($n = 4$), EVT-TS^CT & -TS^blast ($n = 4$), ST-TS^CT & -TS^blast ($n = 4$), CTB^Term ($n = 3$), TS^Term ($n = 3$), EVT-TS^Term ($n = 3$), and ST-TS^Term ($n = 3$) cells and subjected to a PCA. Note that the data from TS^CT and TS^blast cells are combined ($n = 4$) and their derivative EVTs and STBs as well in the Okae's study. **b**, Volcano plots displaying differentially expressed genes between TS^Term cells and their derivative STBs (left) or EVTs (right). Raw $p$-values are presented in the plots. **c**, Pearson correlation coefficients between the gene expression profiles of ITGA6-positive term CTBs, TS^Term, TS^CT, and TS^blast cells and their derivative STBs (ST-TS^CT, ST-TS^blast, and ST-TS^Term) and EVTs (EVT-TS^CT, EVT-TS^blast, and EVT-TS^Term). **d**, Heatmap representation of relative expression (Z-score) of lineage-specific genes across TS^Term, TS^CT, and TS^blast cells and their derivative STBs and EVTs. Of note, additional TS^Term-, ST-TS^Term-, and EVT-TS^Term-specific genes that match the CTB, STB, and EVT lineage-specific genes from 3D-cultured human pre-gastrulation embryos are listed in the lower part of the map. **e** Expression levels of TS, STB, and EVT marker genes in TS^Term, TS^CT, and TS^blast cells and their derivative STBs and EVTs. Data are presented as mean ± SD of independent preparations of the indicated cell types mentioned in (**a**). **f** Functional annotation of DEGs in TS^Term, ST-TS^Term, and EVT-TS^Term cells by ConsensusPathDB. The $q$-values corresponding to the $p$ values after FDR corrections are smaller than 0.05. Source data are provided in the Source Data file.

condition, which effectively inhibits GCM1 activity. As such, combining hypoxia and EGF/CASVY is able to sustain long-term culture of TS^Term cells from ITGA6-positive term CTBs. The TS^Term cells can be propagated to >20 passages and efficiently differentiated into STBs and EVTs. Of note, in the study by Cinkornpumin et al., TS^CT and TS^blast cells were maintained under hypoxic conditions to support self-renewal and inhibit spontaneous differentiation[19]. In a different study, Castel et al. also maintained their TS^naive and iTS cells under hypoxic conditions[18]. We speculated that suppression of residual GCM1 expression by hypoxia may facilitate the maintenance of the aforementioned TS cells. Of the genes that are expressed in the TS^Term and ST-TS^Term cells and the CTB and STB lineages derived from 3D-cultured human blastocysts (Fig. 6d), *TMEM131L* and *RIPOR2* are of interest. TMEM131L has been shown to inhibit the canonical WNT signaling by promoting lysosome-dependent degradation of LRP6 in the regulation of thymocyte proliferation[43]. RIPOR2 also named as PL48 was identified as a novel gene involved in the STB differentiation[44]. These findings suggested possible roles for *TMEM131L* and *RIPOR2* in the control of the self-renewal and differentiation of TS cells. On the other hand, we noticed that the efficiency of in vivo differentiation of TS^CT and TS^blast cells into EVTs in the engraftment study of NOD-SCID mice is relatively low compared with that of TS^Term cells. Whether this discrepancy is due to the residual GCM1 activity in TS^CT and TS^blast cells or the differentially expressed genes between TS^Term, TS^CT, and TS^blast cells awaits further study.

The present study supports a model that the functional antagonism between GCM1 and ΔNp63α dictates the fate of TS cells. In line with this model, meta-analysis of the datasets from single-cell RNA-seq of human first-trimester placentas[45] and RNA-seq of TS^CT and TS^blast cells and their derivative STBs and EVTs[15] showed that expression of *ΔNp63α* and trophoblast stemness genes was mutually exclusive to that of *GCM1* and its target genes associated with trophoblast differentiation in TS cells and differentiated trophoblasts (Supplementary Fig. 9). The intestinal epithelial stem cells migrate up the villus experiencing a decreasing WNT gradient and an increasing BMP gradient, which promote the differentiation of stem cells into different mature cell types[46]. Based upon the anatomical distribution of ΔNp63α-expressing CTBs, ΔNp63α- and GCM1-coexpressing trophoblasts, and GCM1-expressing EVTs in the proximal, mid, and distal regions of cell column (Fig. 1a), it is feasible to speculate that the proliferative ΔNp63α-expressing CTBs are exposed to niche factors that maintain their stemness and later become GCM1-expressing EVTs when they migrate away from the proximal region of the cell column. Identification and characterization of the physiological factors that regulate ΔNp63α or GCM1 activity would give a better understanding of the niche factors that control TS cell maintenance and differentiation.

TS^Term cells are a valuable model for functional characterization of trophoblast differentiation. In this regard, we generated *GCM1*-KO TS^Term cells to show that GCM1 is essential for STB or EVT differentiation. We further identified *CKMT1* as a GCM1 target gene crucial for STB differentiation by gene expression profiling in the STBs derived from wild-type and *GCM1*-KO TS^Term cells. Suppression of CKMT1 activity by RNA interference or cyclo-creatine inhibits trophoblast fusion and hCGβ synthesis, indicating that the CKMT1/creatine phosphate energy shuttle is critical for STB differentiation. Clinically, we showed that decreased CKMT1 expression is associated with PE and more specifically in the pre-eclamptic STBs. The potential of CKMT1 as a diagnostic marker of PE warrants further investigation. Myoblast fusion in myogenesis is energy-dependent and enhanced by creatine treatment in a creatine kinase-dependent manner[47], lending support to the idea that STB differentiation is also an energy-demanding process modulated by the GCM1-CKMT1 axis. Human and mouse trophoblasts with cardiac repair capacity have recently been demonstrated in a mouse model of myocardial infarction (MI)[48,49]. It is intriguing to test the capacity of TS^Term cells in repair of damaged cardiomyocytes in MI mice. Our study indicates that TS^Term cells can easily be established from every individual's placenta and propagated for lifetime storage as an abundant source of stem cells, which may offer huge potential for autologous cell therapy. Finally, it is worth mentioning that choriocarcinoma and non-trophoblast cell lines are used for characterizing the antagonism between ΔNp63α and GCM1 in the present study. Future studies should try to make use of primary trophoblast cells and TS cell lines as much as possible to reinforce the conclusion of the present study.

## Methods

**Human samples and animals**. The use of placental tissues for analysis of GCM1, ΔNp63α, and CKMT1 expression and establishment of TS^Term cells was approved by the Institutional Review Board of MacKay Memorial Hospital (no. 21MMHIS094e and no. 18MMHIS182) and Cathay General Hospital (no. CT-100058), Taiwan. Placental tissues were collected from preterm or term pregnancies, or patients with clinical diagnosis of preeclampsia. The patients with age above 45 years old, fetal anomaly, gestational diabetes mellitus or other medical complications were excluded. All the personal information was delinked from the placental tissue and a waiver of written informed consent from the subjects enrolled was granted by IRB. The experiments were carried out in accordance with the IRB guidelines and regulations and the criteria set by the Declaration of Helsinki. Nine-week-old male NOD-SCID mice were obtained from the National Laboratory Animal Center of Taiwan and maintained under specific-pathogen-free conditions. Protocols and use of animals were undertaken with the approval of the Institutional Animal Care and Use Committee of Academia Sinica (Protocol ID: 19-12-1374).

**Plasmid constructs**. Human ΔNp63α cDNA fragment with a C-terminal FLAG, HA or Myc tag was cloned into pcDNA3.1 (Invitrogen, Carlsbad, CA), pEF1 (Invitrogen), p3XFLAG-CMV-14 (Sigma-Aldrich, St. Louis, MO) or a lentiviral vector containing an expression cassette of puromycin resistance gene or GFP (pCDH or pCDH-GFP, SBI, Mountain View, CA) to generate pΔNp63α-FLAG, pΔNp63α-HA, pΔNp63α-

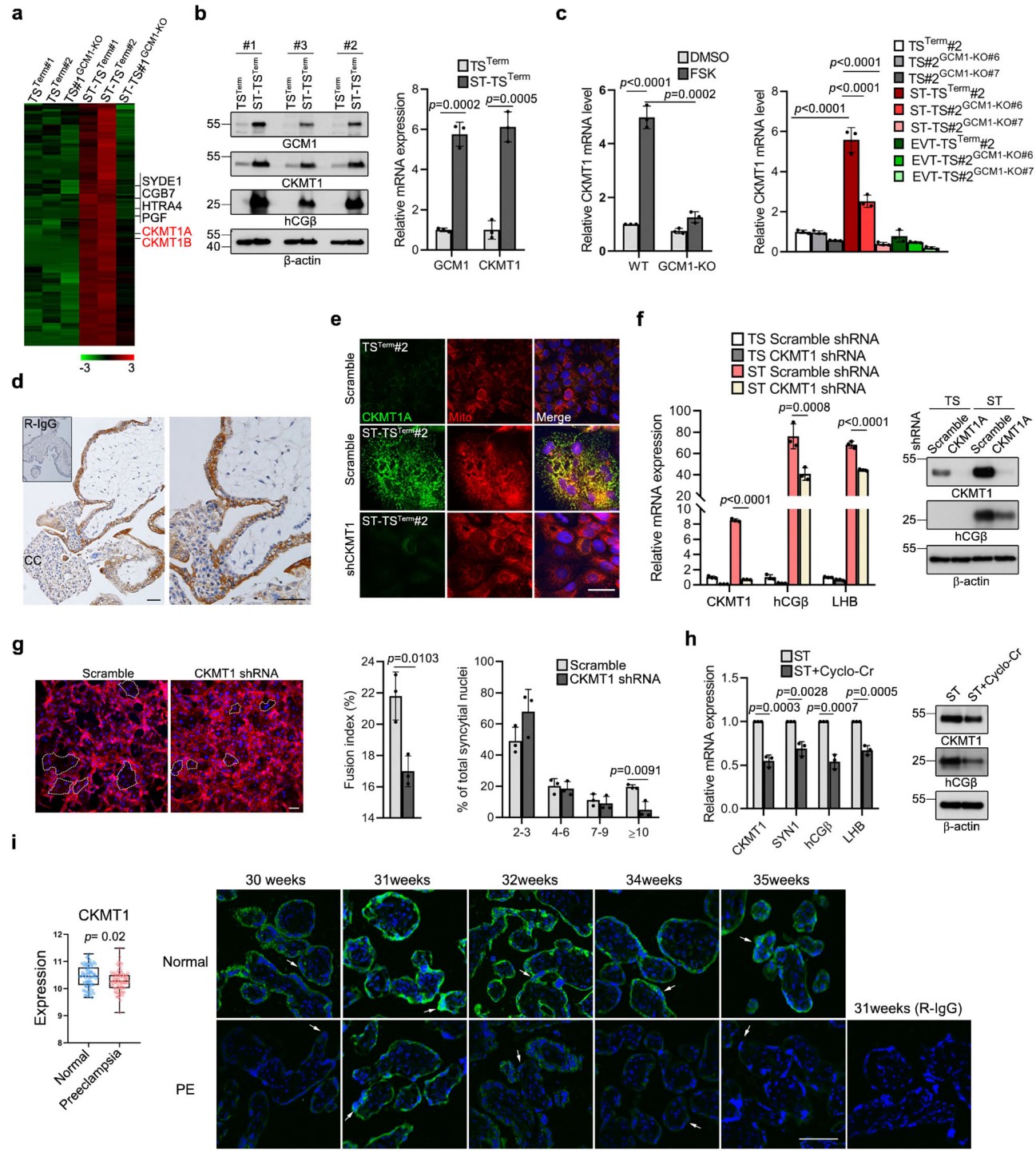

Myc, pCDH-ΔNp63α-FLAG or pCDH-GFP-ΔNp63α-FLAG, respectively. Human OVOL-1 cDNA fragment with a C-terminal FLAG was cloned into pDNA3.1 to generate pOVOL-1-FLAG. The expression plasmid pHA-GCM1 has been described previously[24]. A GCM1 cDNA fragment with a C-terminal HA tag was cloned into pCDH and pCDH-GFP to generate pCDH-GCM1-HA and pCDH-GFP-GCM1-HA plasmids, respectively. The pHTRA4-1Kb and E1bLUCGCM1-2K reporter plasmids harboring human HTRA4 and GCM1 promoters has been described previously[21,28]. Human ΔNp63 genomic fragment containing nucleotides −823 to +262 relative to the transcription start site was cloned into pGL3-basic (Promega, WI, USA) to generate the pGL3-ΔNp63αLuc reporter plasmid. The pGL3-p63bswtLuc and pGL3-p63bsmutLuc reporter plasmids were constructed by cloning into pGL3-basic two copies of WT and mutant p63-binding sites derived from the p63 target gene MTSS1[50]. Lentiviral pLKO.1-Puro short hairpin (sh) RNA expression plasmids for scramble, GCM1, ΔNp63α, GATA3, and CKMT1 were obtained from the National RNAi Core Facility of Taiwan (Taipei, Taiwan). The shRNA sequences are listed in Supplementary Table 1.

**Cell lines, transfection, and lentivirus transduction**. The following cell lines were obtained from ATCC: 293T (Cat# CRL-3216), BeWo (Cat# CCL-98), JAR (Cat# HTB-144), JEG3 (Cat# HTB-36), and the p53-deficient Hep3B (Cat# HB-8064). TS^Term#1, TS^Term#2, and TS^Term#3 were established from term CTBs in this study. The GCM1 gene in JEG3, TS^Term#1, and TS^Term#2 cells was knocked out by the CRISPR/Cas9 system to generate the JEG3GCM1-KO, TS#1^GCM1-KO, TS#2^GCM1-KO#6, and TS#2^GCM1-KO#7 cells, respectively. 293T cells were maintained at 37 °C in DMEM supplemented with 10% fetal bovine serum (FBS) and 1% penicillin/streptomycin/glutamine. JEG3, JEG3GCM1-KO, and Hep3B cells were maintained at 37 °C in MEM supplemented with the 10% FBS and 1% penicillin/streptomycin. BeWo and JAR cells were maintained at 37 °C in F12K supplemented with 15% FBS and 1% penicillin/streptomycin/glutamine. Maintenance of TS^Term and GCM1-KO TS^Term cells were described in the Trophoblast stem cells and trophoblast differentiation subsection. For transient expression, cells were transfected with the indicated reporter and expression plasmids using the Lipofectamine 2000 reagent (Invitrogen). For luciferase assays, cells were harvested

**Fig. 7 Regulation of STB differentiation by the GCM1-CKMT1 axis. a** *CKMT1A* and *-1B* are GCM1 target genes in STB differentiation. **b** *CKMT1* expression is upregulated in the STBs differentiated from TS$^{Term}$#1, -2, and -3 cells, respectively. **c** Upregulation of *CKMT1* expression in the FSK-induced STB differentiation is impaired in JEG3 (left) and TS$^{Term}$ (right) cells by *GCM1* knockout. **d** Expression of CKMT1 in first-trimester STBs (brown). The inset shows a section stained with normal R-IgG. Image of higher magnification is presented (right). CC, cell column. Scale bar, 50 μm. **e** Subcellular localization of CKMT1 in the mitochondria of STBs. Scramble control TS$^{Term}$#2 and ST-TS$^{Term}$#2 cells and *CKMT1*-knockdown ST-TS$^{Term}$#2 cells were stained with MitoTracker Red (Mito) and CKMT1 Ab. Scale bar, 50 μm. **f, g** STB differentiation is compromised by *CKMT1* knockdown. Expression of STB markers *hCGβ* and *LHB* (**f**) and cell fusion (**g**, E-cad immunostaining) are decreased in *CKMT1*-knockdown ST-TS$^{Term}$#2 cells. Syncytial margins are marked with a stippled line. Cell fusion efficiency was measured by fusion index or the distribution of syncytia containing varying numbers of nuclei. Scale bar, 100 μm. **h** Regulation of STB differentiation by the creatine phosphate-CKMT1 system. TS$^{Term}$#2 cells were induced into STBs by FSK in the presence or absence of cyclocreatine (Cyclo-Cr) for 72 h, followed by analysis of CKMT1, SYN1, hCGβ, and LHB transcripts or proteins. Data are presented as mean ± SD of independent experiments (*n* = 3 in **b**, **c**, **f**, **g**, **h**). Differences were assessed by unpaired two-tailed Student's *t*-test (**b**, **c** (left), **g**, **h**) or two-way ANOVA with Tukey's post hoc test (**c** (right), **f**). **i** Decreased *CKMT1* expression is associated with preeclampsia (PE). Meta-analysis of the GSE75010 microarray dataset of PE (*n* = 80) and normal control (*n* = 77) women for *CKMT1* expression (left). The center dashed line denotes the median value, while the box contains the 25th to 75th percentiles of the dataset. The whiskers mark the minimum and maximum values of the dataset. Differences were assessed by the Mann Whitney test. Immunofluorescence microscopy of CKMT1 (green) in normal and PE placentas at different weeks of gestation (right). White arrows point to CKMT1 expression in the STB of normal and PE placentas. Scale bar, 100 μm. Source data are provided in the Source Data file.

in the reporter lysis buffer (Promega) 48 h post-transfection. The reaction was initiated by adding luciferin, and the light emission was monitored by a luminometer (Perkin Elmer 1420 Victor Light luminescence counter, Waltham, MA, USA). Specific luciferase activities were normalized by protein concentration. Protein concentrations were measured by using Pierce BCA protein assay kit (Thermo Fisher Scientific, Waltham, MA, USA). For stable expression of exogenous ΔNp63α-FLAG or GCM1-HA, cells were infected with recombinant lentivirus strains harboring pCDH-ΔNp63α-FLAG, pCDH-GFP-ΔNp63α-FLAG, pCDH-GCM1-HA or pCDH-GFP-GCM1-HA. To establish scramble control, ΔNp63α-, GCM1- or CKMT1-knockdown cells, cells were infected with recombinant lentivirus strains harboring a scrambled, ΔNp63α, GCM1 or CKMT1 shRNA, respectively. The infected cells were subjected to antibiotic selection using 5 μg/ml of puromycin or flow cytometry, and the puromycin-resistant clones or GFP-positive cells were pooled for studies.

**Antibodies**. Homemade and commercially available antibodies were used for immunoblotting (IB), immunofluorescence staining (IF), immunohistochemistry (IHC), chromatin immunoprecipitation (ChIP), in situ proximity ligation assay (PLA), and flow cytometry (Flow) at the indicated dilutions. Homemade antibodies include anti-GCM1 rabbit Ab for IB 1:1000; IF 1:100; IHC 1:100; PLA 1:100, anti-GCM1 guinea pig Ab for IB 1:1000; IF 1:100, and anti-HTRA4 rabbit Ab for IB: 1:1000. Antibodies obtained from the commercial source include anti-p63α rabbit mAb (Cell Signaling Cat# 13109) for IF 1:400; ChIP 1:100, anti-p63 mouse mAb clone 4A4 (Abcam Cat# ab735) for IB 1:1000; IHC 1:100; PLA 1:50, anti-hCGβ rabbit Ab (Dako Cat# A0231) for IB 1:2000, anti-hCGβ mouse mAb (Santa Cruz Cat# sc-271062) for IB 1:2000; IF 1:100, anti-E-Cadherin mouse mAb (BD Biosciences Cat# 610182) for IF 1:200, anti-CK7 mouse mAb (Abcam Cat# ab9021) IHC 1:200, anti-ITGA6 rat mAb (Santa Cruz Cat# sc-19622) for Flow 1:100, anti-HLA-G mouse mAb (Abcam Cat# ab7758) for Flow 1:100, anti-HLA-G mouse mAb (Abcam Cat# ab7759) for IF 1:100, anti-HLA-G mouse mAb (Santa Cruz Cat# sc-21799) for IHC 1:50, anti-RACK1 mouse mAb clone B-3 (Santa Cruz Cat# sc-17754) for IB 1:1000, anti-GATA3 mouse mAb (Santa Cruz Cat# sc-268) for IB 1:1000; IF 1:100, anti-TFAP2C mouse mAb (Santa Cruz Cat# sc-12762) for IF 1:100, anti-Ki67 rabbit Ab (Abcam Cat# ab66155) for IF:200, anti-EpCAM mouse mAb (Cell Signaling Cat# 2929) for IF 1:500, anti-CKMT1A rabbit Ab (Proteintech Cat# 15346-1-AP) for IB 1:1000; IF 1:200; IHC 1:200, anti-FLAG M2-peroxidase mouse mAb (Sigma-Aldrich Cat# A8592) for IB 1:5000, anti-mouse Alexa 568 (Thermo Fisher Scientific Cat# A-11031) for IF 1:300, anti-guinea pig Alexa 568 (Thermo Fisher Scientific Cat# A-11075) for IF 1:300, anti-rabbit Alexa 488 (Thermo Fisher Scientific Cat# A-11034) for IF 1:300, anti-mouse Alexa 633 (Thermo Fisher Scientific Cat# A-21052) for IF 1:300, anti-mouse Alexa 488 (Thermo Fisher Scientific Cat# A-11029) for IF 1:300; Flow 1:100, anti-rat Alexa 568 (Thermo Fisher Scientific Cat# A-11077) for Flow 1:100, MaxFluor 488 secondary Ab for mouse IgG (MaxVision Biosciences Cat# DSMR-H1) for IF 1:1, and MaxFluor 550 secondary Ab for rabbit IgG (MaxVision Biosciences Cat# DSMR-H1) for IF 1:1.

**Trophoblast stem cells and trophoblast differentiation**. Placental tissues were collected from healthy women undergoing elective termination of pregnancy or caesarean section. To purify ITGA6-positive CTBs, villous tissues of term placenta were collected, trypsinized, and subjected to Percoll gradient centrifugation to enrich trophoblasts, which were further sorted out by flow cytometry using ITGA6 Ab and Alexa Fluor 568-conjugated secondary Ab in a BD FACSAria IIIu sorter (FACSDiva software version 8.0.1, BD Biosciences, San Jose, CA). The ITGA6-positive CTBs were seeded onto culture plates pre-coated with 5 μg/ml Col IV and incubated at 37 °C under hypoxia (1% O$_2$, 5% CO$_2$, and 94% N$_2$ in a multigas incubator (Astec, Fukuoka, Japan)) in complete TS cell medium[15], which is DMEM/F12 supplemented with 0.1 mM 2-

mercaptoethanol, 0.2% FBS, 0.3% BSA, 0.5% Penicillin-Streptomycin, 1% ITS-X supplement, 1.5 μg/ml L-ascorbic acid, and 50 ng/ml EGF plus small molecules 5 μM CHIR99021, 0.5 μM A83-01, 1 μM SB431542, 0.8 mM VPA, and 5 μM Y27632 (CASVY). Highly proliferative TS$^{Term}$ cells were established after three or four passages and frozen at early passages or subjected to analysis of GCM1 and ΔNp63α expression and differentiation assays. For induction of EVT-TS$^{Term}$ cells, TS$^{Term}$ cells were seeded onto culture plates pre-coated with 1 μg/ml Col IV and incubated at 37 °C under normoxia (21% O$_2$, 5% CO$_2$, and 74% N$_2$) for 96 h in EVT medium, which is DMEM/F12 supplemented with 0.1 mM 2-mercaptoethanol, 0.5% Penicillin-Streptomycin, 0.3% BSA, 1% ITS-X supplement, 100 ng/ml neuregulin 1 (NRG1), 7.5 μM A83-01, 2.5 μM Y27632, and 4% Knockout serum replacement (KSR). Subsequently, the medium was replaced with the EVT medium without NRG1, and Matrigel was added to a final concentration of 0.5% for additional 4–10 days. For induction of ST-TS$^{Term}$ cells, TS$^{Term}$ cells were cultured in ST medium, which is DMEM/F12 supplemented with 0.1 mM 2-mercaptoethanol, 0.5% Penicillin–Streptomycin, 0.3% BSA, 1% ITS-X supplement, 2.5 μM Y27632, 5 μM forskolin, and 4% KSR, at 37 °C under normoxia for 3–4 days. To study the effect of EGF/CASVY on trophoblast differentiation, BeWo cells were cultured in complete TS medium or incomplete TS medium (without EGF/CASVY) for 24 h. Cells were then harvested for quantitative RT-PCR analysis of ΔNp63α, TEAD4, EPCAM, ELF5, GCM1 and its target genes. GCM1-knockout JEG3 and TS$^{Term}$ cells were generated by the CRISPR/Cas9 system. In brief, JEG3 or TS$^{Term}$ cells were infected with lentiviruses harboring the pAll-Cas9.Ppuro vector (provided by the National RNAi Core Facility of Taiwan) with a sgRNA sequence (5′-CAG-GAAGGCGTCCAATTGCC-3′) targeting exon 6 of the human GCM1 gene. After puromycin selection, the surviving cells were seeded individually into 96 wells and genomic DNA of each single colony was extracted for PCR amplification and sequencing of the sgRNA targeting site.

**Immunohistochemistry and RNAscope in situ hybridization**. First-trimester and full-term human placental tissues were fixed in 10% neutral buffered formalin (approximately 4% formaldehyde), dehydrated, embedded in paraffin, and sectioned at 5 μm. The sections were deparaffinized, rehydrated, and incubated with IgG, rabbit anti-GCM1 Ab or ΔNp63 mAb (Abcam, Cambridge, UK) and then MaxFluor 488 secondary Ab for mouse IgG and MaxFluor 550 secondary Ab for rabbit IgG according to the manufacturer's instructions (MaxVision Biosciences, Kenmore, WA). Nuclei were stained with DAPI. Immunofluorescence was examined under an Olympus laser scanning confocal microscope (FV3000) (Shinjuku, Tokyo, Japan). Images were prepared for presentation using Adobe Photoshop CS6 v13.0. Expression of trophoblast stemness- and differentiation-related genes was assessed in TS$^{Term}$ cells or their derivative STBs and EVTs by immunofluorescence microscopy using GCM1, hCGβ, HLA-G, p63α, GATA3, TFAP2C, and MKI67 primary Abs; and AlexaFluor 488, AlexaFluor 633, and AlexaFluor 568 secondary Abs. To examine CKMT1 expression in placenta, paraffin-embedded first-trimester placental tissue sections were subjected to chromogenic staining using CKMT1A Ab (Proteintech, Rosemont, IL), HRP-conjugated secondary Ab, and DAB chromogen (Vector Labs, Burlingame, CA). Subcellular localization of CKMT1 was studied in the undifferentiated scramble control TS$^{Term}$ cells and the STBs derived from scramble control and CKMT1-knockdown TS$^{Term}$#2 cells using MitoTracker Red (Cell Signaling, Danvers, MA) and CKMT1 Ab for costaining of mitochondria and CKMT1, respectively. In addition, tissues of normal and preeclamptic placentas of different gestational ages were snapped in liquid nitrogen for cryosectioning. Sections were stained with CKMT1A Ab, followed by incubation of AlexaFluor 488-conjugated secondary Ab for immunofluorescence microscopy. Expression of ΔNp63α and GCM1 transcripts in placental trophoblasts were measured by RNAscope in situ hybridization. First-trimester and full-term human placental tissue sections were deparaffinized, rehydrated, and incubated with GCM1 and ΔNp63α probes according to the

manufacturer's instructions (Advanced Cell Diagnostics, Newark, CA). Signals were visualized using HRP-based Green and AP-based Fast Red chromogens and examined under an Olympus BX51 microscope.

**Cell fusion assay**. For cell fusion analysis, scramble or ΔNp63α shRNA-expressing JEG3 cells treated with or without 50 μg/ml FSK for 48 h, followed by immuno-fluorescence staining with E-cadherin Ab (E-cad, BD Biosciences) and AlexaFluor 568-conjugated secondary Ab. Similar approach was used to study cell fusion in the STBs derived from wild-type, *GCM1*-KO, ΔNp63α-FLAG-expressing, and *CKMT1*-knock-down TS<sup>Term</sup> cells. Images were captured by the aforementioned confocal microscope. Three microscopic fields per sample were randomly selected for examination in each of three independent experiments. Quantification of cell fusion was calculated as a fusion index of (N-S)/T, where N is the number of nuclei in the syncytia, S is the number of syncytia, and T is the total number of nuclei counted. In addition, cell fusion efficiency was measured by distribution of syncytia of different sizes (containing varying numbers of nuclei) as a ratio of the total number of nuclei per syncytium size over the total number of syncytial nuclei counted.

**Coimmunoprecipitation and in situ proximity ligation assay (PLA)**. To study the interaction between GCM1 and ΔNp63α or OVOL1, 293 T cells were transfected with pHA-GCM1 and pΔNp63α-FLAG or pOVOL1-FLAG. At 48 h post-transfection, cells were harvested in lysis buffer containing 50 mM Tris-HCl (pH 8.0), 150 mM NaCl, 2 mM EDTA, 10% glycerol, 0.5% NP-40, 1 mM DTT, 5 mM NaF, 1 mM Na$_3$VO$_4$, 1 mM PMSF, and a protease inhibitor cocktail (Sigma-Aldrich), followed by consecutive immunoprecipitation and immunoblotting with FLAG and HA (Sigma-Aldrich) mAbs. Immunoblot band intensities were quantified using ImageJ densitometry software (v 1.53k). Interaction between endogenous ΔNp63α and GCM1 in TS<sup>Term</sup> cells was visualized using a Duolink PLA kit (Sigma-Aldrich) according to the manufacturer's instructions. In brief, TS<sup>Term</sup> cells were cultured under normoxic conditions for 96 h to increase the number of both ΔNp63α- and GCM1-positive cells. Subsequently, cells were incubated with different combinations of IgG, rabbit anti-GCM1 Ab, and ΔNp63 mAb and the PLA probes, Duolink PLA probe anti-mouse minus and Duolink PLA probe anti-rabbit plus. The samples were then processed for ligation and amplification, mounted using a DAPI-containing mounting medium, and examined under a confocal microscope. Five microscopic fields per sample were randomly selected for examination in each of three independent experiments. Quantification of PLA signal per nucleus was calculated as total PLA puncta divided by the number of all nuclei in the field.

**Quantitative RT-PCR**. To study the regulation of trophoblast stemness and dif-ferentiation genes by GCM1 and ΔNp63α, BeWo cells stably expressing ΔNp63α-FLAG or GCM1 shRNA and JEG3 cells stably expressing GCM1-HA or ΔNp63α shRNA were mock treated or with 50 μg/ml FSK or 1 mM DB-cAMP for 24 h and then harvested for RNA isolation using RNeasy Mini kit (Qiagen, Hilden, Germany). The isolated RNA was transcribed into cDNA using SuperScript III reagents (Invitrogen) with an oligo-(dT)$_{20}$ primer. Quantification of the transcript levels of indicated genes was performed in the LightCycler 480 system (software v1.5.1.62, Roche, Basal, Switzerland) using a commercial SYBR Green reaction reagent (Qiagen) and specific primer sets. The sequences of the primer sets were listed in Supplementary Table 1.

**ΔNp63α oligomerization and degradation**. To study the effect of GCM1 on ΔNp63α oligomerization, 293T cells were transfected with pΔNp63α-Myc and pΔNp63α-FLAG plus increasing amounts of pHA-GCM1. At 48 h post-transfec-tion, cells were harvested for coimmunoprecipitation analysis with HA, FLAG, and Myc mAbs. To study the role of GCM1 in the FSK-induced downregulation of ΔNp63α, WT and *GCM1*-KO JEG3 cell were treated with 50 μM FSK alone or plus 20 μM MG132 for 18 h. Cells were then harvested for quantitative RT-PCR or immunoblotting analysis of ΔNp63α, GCM1, HTRA4 or hCGβ.

**RNA sequencing**. WT and *GCM1*-KO TS<sup>Term</sup> cells and their derivative STBs and EVTs were harvested for RNA purification using the RNeasy min kit and RNase-free DNase (Qiagen, Hilden, Germany). To assess the RNA integrity, RNA integrity number (RIN) was created using RNA 6000 Nano and 2100 Bioanalyzer System (Agilent Technologies, Santa Clara, CA). Each sample had an RIN (RNA integrity number) value above 7. RNA-seq libraries were prepared using Universal RNA-Seq with NuQuant Kit (Tecan Genomics, Redwood City, CA) according to the man-ufacturer's instructions. The libraries were sequenced on the NovaSeq 6000 plat-form (Illumina) to produce 45–49 million 2 × 150 bp paired-end reads per sample. RNA-seq reads were trimmed using CLC Genomics Workbench v10 to a minimum quality score of 0.01 (equivalent to Phred score of 20), and adaptors were also removed. The trimmed reads were aligned to the reference genome Homo sapiens GRChg38 using CLC Genomics Workbench v10. Gene expression was measured by FPKM (fragments per kilobase of transcript per million mapped reads) with Subread package (featureCounts, v1.6.5). Differentially expressed genes were identified using DESeq2[51] (fold change >3 and *p* < 0.05 in Fig. 6 or fold change >8 and *p* < 0.01 in Fig. 7a). For principal component analysis (PCA), the expressed genes of the RNA-seq datasets of interest were standardized to a Z-score. A three-dimensional PCA (performed with MATLAB version R2015a) was applied to explore the patterns of gene expression variation across various placental cell types.

Functional annotation of the lineage-specific genes was performed using ConsensusPathDB[52].

**Engraftment of TS<sup>Term</sup> and *GCM1*-KO TS<sup>Term</sup> cells into NOD-SCID mice**. TS<sup>Term</sup>#2 and TS#2<sup>GCM1-KO#6</sup> cells were grown to 80-90% confluence in complete TS medium and dissociated with TrypLE. Approximately 5 × 10$^6$ TS<sup>Term</sup>#2 or TS#2<sup>GCM1-KO#6</sup> cells in 200 μL of a 1:2 mixture of Matrigel and DMEM/F12 containing 0.3% BSA and 1% ITS-X supplement were subcutaneously injected into 9-week-old male NOD-SCID mice. The mice were kept at 21 ± 2 °C at 59 ± 1% humidity. The dark/light cycle was 12 h each, with light hours between 7:00 am and 7:00 pm. Lesions were collected 10 days after injection, fixed with formalin, and embedded in paraffin wax for immunohistochemistry of CK7, hCGβ, and HLA-G.

**Statistics and reproducibility**. Differences were assessed by unpaired two-tailed Student's *t*-test, Mann Whitney test, and one- or two-way ANOVA with Tukey's post hoc test using GraphPad Prism 8 software (GraphPad Software, San Diego, CA). Data with error bar are presented as Means and standard deviations in the figures of this study. A *p* value of <0.05 was considered statistically significant. Similar results were obtained in Figs. 1b–h, 2a, c, e, f, 3a, 4e, h, and 7b, f, h and Supplementary Figs. 1c–e, 3d, f, 4, and 7a in three independent experiments and representative images are shown.

**Reporting summary**. Further information on research design is available in the Nature Research Reporting Summary linked to this article.

## Data availability
The RNA-seq and ChIP-chip data generated in this study have been deposited in the Gene Expression Omnibus (GEO) database under accession codes GSE158901 and GSE158894. The human reference genome assembly GRChg38R [https://www.ncbi.nlm.nih.gov/assembly/GCF_000001405.26/] was used for alignment of the RNA-seq reads of TS<sup>Term</sup> cells and their STB and EVT derivatives. The RNA-seq datasets of TS<sup>CT</sup> and TS<sup>blast</sup> cells (JGAD000073 and JGAD000115) and the single-cell RNA-seq datasets of 3D-cultured human blastocysts (GSE136447) were used for comparative analysis of gene expression profiles in TS<sup>Term</sup> cells and their STB and EVT derivatives. Meta-analysis of *GCM1* and *ΔNp63α* gene expression in TS<sup>CT</sup> and TS<sup>blast</sup> cells and differentiated trophoblasts was performed on the JGAD000073 and JGAD000115 datasets and the single-cell RNA-seq datasets of human first-trimester trophoblasts (GSE89497). Raw data of Figs. 1b–i, 2, 3a, d, 4d–h, 6a, e–f, and 7b, c, f–i and Supplementary Figs. 1c–f, 2, 3b–f, 4, 5, and 7 are provided as a Source Data file. Additional data are available from the corresponding author upon reasonable request. Source data are provided with this paper.

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

## Acknowledgements

The authors would like to thank Dr. Tso-Pang Yao for constructive criticism of the manuscript, the Data Science Statistical Cooperation Center of Academia Sinica (AS-CFII-108-117) for statistical support, and Mr. Tzu-Chi Chuang for assistance in graphical presentation. This work was supported by grants (to H.C.) from Ministry of Science and Technology (grant number 109-2311-B-001-011-MY3), the National Health Research Institutes (grant number NHRI-EX110-11032SI), and Academia Sinica, Taiwan.

## Author contributions

L.-J.W. designed the study and performed the experiments and data analysis and interpretation. C.-P.C., Y.-S.L., G.-D.C., and M.-L.C. contributed reagents and performed bioinformatics analysis of the RNA-seq and ChIP-chip data. P.-S.N., Y.-H.P., H.-F.L., and C.-H.P. performed the experiments and data analysis and interpretation. H.C. conceptualized and designed the study, performed analysis and interpretation of data, and wrote the manuscript.

## Competing interests

The authors declare no competing interests.
