## [Peer Review File · Nature Communications]

Reviewers' Comments:

Reviewer #1:

Remarks to the Author:

The authors describe a mechanism by which trophoblast stem (TS) cells may be maintained in the human placenta. Specifically, they claim that p63 (specifically the deltaNp63alpha isoform) maintains stemness and GCM1 promotes differentiation and the functional relationship between the two proteins determines the balance between these two states. Human TSC derivation has previously been achieved using culture additives, which in this manuscript are termed EGF/CASVY; however, the mechanism by which this culture condition maintains TSC has not been determined. The authors of this manuscript claim that EGF/CASVY activates deltaNp63alpha, thus suppressing GCM1 activity, and promoting TS "conversion" from BeWo choriocarcinoma cells as well as cytotrophoblast (CTB) derived from term placenta. However, term CTB-derived TSC using EGF/CASVY is only stably achieved under hypoxic conditions (1% oxygen) where deltaNp63alpha is further upregulated and GCM1 activity is further downregulated. Further, the authors also point to CKMT1, a GCM1 target, as crucial for syncytiotrophoblast (STB) differentiation and a protein whose expression is decreased in patients with preeclampsia.

Overall there is a significant amount of data to support a ying-yang relationship between p63 and GCM1. However, the signaling relationships are more complicated than claimed in the paper. Furthermore, the authors do not in fact establish that the mechanism of EGF/CASVY media can be attributed to the p63-GCM1 relationship, nor do they establish that the ability to derive TSC from term CTB (which has failed in the past) is in fact due to suppression of GCM1 expression by hypoxia. In addition, the in vivo tumor formation assay data as well as RNAseq data showing that term CTB have in fact been converted to TSC are at best weak. Nevertheless, the protocol for derivation of TSC from term CTB would be a valuable tool for the field. Further, the relationship between GCM1-CKMT1 and STB differentiation and preeclampsia is also novel and significant. Specific points (both major and minor) are listed below, which, if addressed, would help align the conclusions with the results.

Major points:

1) The phrase "TS conversion" is a bit problematic, particularly in context of the application of EGF-CASVY media to blastocyst and first trimester placental tissue for derivation of TSC. It makes more sense when applied to pluripotent stem cells (that these cells are being "converted" to TSC); but in context of tissues, the media is likely not "converting" one cell type into another, but rather capturing a subpopulation of CTB in a particular stem-like state.

2) While the physical interaction and reciprocal expression between p63 and GCM1 seems sound, the relationship is only characterized for GCM1 regulation of p63. The authors do acknowledge the in the text, but the title implies two-way regulation.

3) The authors claim that p63 inhibits GCM1 function and not GCM1 expression. This conclusion was likely reached due to the binding at the transactivation domain and the unchanged expression of GCM1 upon manipulation of p63 expression. However, the readout of GCM1 function/activity was the activation of differentiation genes. To be able to make this conclusion, the authors would need to show reduced GCM1 binding by ChIPseq in cell lines overexpressing p63. Alternatively, increasing amounts of GCM1 in the presence of overexpressed p63 should "rescue" expression of HTRA4 and hCGbeta.

4) The authors conclude "...EGF/CASVY induces dedifferentiation of BeWo cells by activation of deltaNp63alpha and trophoblast stemness gene expression, and thereby suppress GCM1 activity and differentiation-related target gene expression." The authors do show EGF/CASVY induces dedifferentiation and increases p63 and trophoblast stemness genes. However, this can only be concluded if EGF/CASVY does not induce stemness genes in the absence of p63.

5) Similarly, hypoxia is concluded to downregulate GCM1 promoter activity (and therefore GCM1 expression). However, the details of hypoxia treatment are neither described, nor confirmed (i.e. by evaluation of HIF components). Also, hypoxia cannot necessarily be deemed causative here, unless GCM1 overexpression in hypoxic conditions reverses the increase in p63 and decrease in hCGbeta.

6) Many of the experiments are done using choriocarcinoma cells, which are known to be suboptimal models of in vivo human trophoblast. For this reason, at the very least, all figures should be properly labeled to indicate the cell type being used. It might best, however, to repeat

at least some of these experiments in TS-term cells.

7) lines 74-76: The meaning and relevance of this sentence are unclear: JEG3 are choriocarcinoma cells, with abnormal ploidy and inability to make syncytiotrophoblast. Their ability to proliferate and invade is a property of their origin from a tumor; that they express CDX2, ELF5, and C19MC is because they originate from a choriocarcinoma. To say that they may be better than BMP4-treated hESC as a model for bona fide human TS because of expression of a handful of markers is disingenuous at best.

8) Lines 113-116: It is misleading to state that the role of p63 in TS maintenance has not been previously addressed (see References #26 and 27, as well as Lee et al. Human Pathology, 2007).

9) A lot of extrapolation is made from mouse TSC to human TSC, despite extensive data supporting significant differences between these two species. Importantly, EOMES has not been identified as a TSC-associated gene in the human placenta, and has in fact been found to be absent in both human blastocyst (Blakeley et al. 2015) and early gestation trophoblast (Soncin et al. 2018). It should be kept in mind that JEG3 and BeWo cells are choriocarcinoma cells and as such likely express many genes NOT found in primary human TSC.

10) Figure 1h: Not sure I see the "effect of FSK on Δ Np63 α expression was abrogated by GCM1 knockdown" since (at least by eye) the downregulation of p63 following FSK treatment is similar with either scramble or GCM1 shRNA. It might help if the authors showed a quantification of the western blot (i.e. underneath the blot). However, it also appears that at the RNA level, there isn't any significant difference in p63 expression under these different conditions.

11) Figure 2b: If the authors are trying to implicate a role for p63 in indirect inhibition of GCM1 activity, they would need to knockdown GCM1 in the context of p63 knockdown and show that the HTRA4 promoter activity being measured here is in fact GCM1-dependent.

12) Continuous passage under hypoxia (1% oxygen) is a very stringent oxygen tension for cellular maintenance and in fact has been shown to induce significant metabolic stress, even in trophoblast. This needs to at least be discussed, as a potential issue for longterm maintenance of TS-term cells. In addition, the actual method for maintaining hypoxic culture is not explained in detail: what type of incubator was used, and were cells maintained under these conditions even during feeding and passage? In Figure 4e, when TS-term are shown under "normoxia," were these cells temporarily exposed to higher oxygen or actually passaged under normoxia?

13) The authors emphasize reciprocal expression of p63 and GCM1, but Figure 4g in fact does not show suppression of GCM1 expression following forced expression of p63 in TS-term cells, even following FSK treatment. How can this be explained?

14) Supplementary Figure 2b: It is not clear that the lone HLAG+/CK7+ cell is in fact part of the tumor formed by the TS-term cells. In the absence of such cells within the tumor, it cannot be said that these cells are in fact bipotential in vivo.

15) The authors claim that they have generated cells similar to TSC derived from first trimester and blastocysts, yet the RNAseq comparison does not include any other cell type (including the cell of origin, term CTB). The authors need to show that in fact TS-term move away from being term-CTB-like, and toward a TS-CT/TS-blast-like state.

16) lines 384-386: Reference #36 shows ELF5 expression in early placenta but does not provide evidence for a role for ELF5 in "establishment" of the TS compartment. This reference also does not discuss EOMES.

17) lines 397-400: that EGF/CASVY may stimulate p63 expression is not supported by data and is therefore highly speculative.

18) Line 444 ("lifetime storage"): Did the authors characterize the ability to freeze-thaw these TS-term cells and still maintain bipotency?

Minor points:

1) On all immunoblots for p63, it would be clearer if the arrow always pointed to the specific band.

2) Is Figure 1e the same as Supplementary Figure 1d?

3) Line 229 – data not shown, would be helpful if at least the assay is described for this conclusion.

4) Line 280: "suppressive effects of hypoxia on GCM1 'activity'" the data describes effects on GCM1 expression. The distinction between "activity" and "expression" is especially important to avoid confusion after the results in the previous section.

5) Figure 2e. This blot does not seem to have the nonspecific band for Δ Np63 α ? Or is that the specific band? It would be good to consistently provide the arrow to the specific band.

6) Figure 2f. Same as figure 2e, the arrow and molecular weights are missing for p63.

7) Antibody catalogue #s and dilutions or concentrations should be provided.

Reviewer #2:

Remarks to the Author:

The manuscript by Wang et al. described an interesting mechanism between GCM1 and Δ NP63 that dictates self-renewing stem state vs. differentiation of human trophoblast stem cells (Human TSC). Authors also used this mechanistic platform to develop a strategy to successfully derive Human TSCs from term placenta, which is still elusive. The biggest strength of this paper that could significantly advance the field is the successful establishment of human TSCs from term placenta. Thus, this reviewer was very interested with the manuscript. However, the manuscript has numerous problems, contradictory data with the published papers, and most importantly insufficient data to define the core hypothesis that GCM1 and Δ NP63 antagonism is the dictating factor for TSC stemness vs. differentiation. My major concerns are mentioned below.

1. I have a major concern about the data, which is shown in Fig. 1a about GCM1 expression in a first-trimester placenta. In contrary to data shown in Fig.1a, earlier publication (Chiu and Chen, *Sci. Rep* 2016 Feb 22;6:21630. doi: 10.1038/srep21630.) reported that GCM1 is expressed in undifferentiated cytotrophoblasts (CTBs, both villous and column CTBs). A quick look of our own single cell RNA-seq data in human first-trimester placentae shows that GCM1 mRNA is expressed in undifferentiated CTBs, although the expression is induced during differentiation. Also, the supplementary Fig. S1a clearly shows GCM1 expression in undifferentiated CTBs. Thus, the data about GCM1 expression, shown in Fig. 1a, does not match with the other observations, which is a major concern. If GCM1 is expressed in undifferentiated CTBs, the whole hypothesis that suppression of GCM1 is important for self-renewing stem state needs to be revisited. Both immunostaining and in-situ hybridization data with multiple placental sections showing both floating and anchoring villi should be included.
2. Why the co-immunoprecipitation experiments are performed with ectopic overexpression system in HEK 293 cells? What are the ectopic protein expression levels compared to endogenous expression levels in trophoblast cells? As authors have good antibodies, they should test it in term CTBs.
3. The JEG3 cell line is choriocarcinoma cell line. It is not a true representative of primary trophoblast cells. Thus molecular mechanism in that context may not be definitive for primary CTBs.
4. Surprisingly, different experiments with JEG3 cells in Fig. 1 show different levels and banding patterns of same protein. For example in Fig. 1C, GCM1 protein is readily detectable in JEG3 cells without FSK treatment. However, the presented data in Fig. 1h shows almost undetectable GCM1 protein under same experimental condition.. The western blot band of Δ NP63 is detected as a single band in some experiments (Fig. 1C, 1e, 1F) and as a double band in other experiments (Fig. 1G, 1H).
5. The GCM1 knockdown efficiency in FSK treated JEG3 cells (Fig. 1H) and corresponding change in Δ NP63 protein is minimal. Thus, the claim in lines 185 and 186 that GCM1 suppresses Δ NP63 activity based on that data is surprising and not conclusive. Also, any change in expression does not mean GCM1 regulates activity of Δ NP63.
6. The increase in differentiation efficiency (Fusion index) of JEG3 cells with FSK is only ~15%. This is a very narrow window to generate any conclusive data. Authors should try induction and repression of cell differentiation upon GCM1 and Δ NP63 overexpression with actual human TSCs.
7. There is no real data showing GATA3 expression is suppressed by Δ NP63. The actual reference indicated regulation of GATA3 by the other isoform TP63 (the whole protein) in hair follicle cells. Thus, direct regulation of GATA3 by Δ NP63 is purely speculative.
8. Also, according to the manuscript GATA3 is a negative regulator of GCM1. However, GATA3 and GCM1 both are abundantly expressed and functions in primary differentiated STs of actual human placenta. Thus, the conclusion drawn from studies in choriocarcinoma cell lines does not fit with actual placenta.

9. As I mentioned earlier, the major strength of this manuscript is the successful derivation of Human TSC lines from term placenta. However, the GCM1 antagonism mechanism, which is indicated in this paper, is surprising. Because, the human TSC lines derived from first-trimester human placenta and reported by (Okae et. al., 2018, Cell Stem Cell) expresses significant amount of GCM1 mRNA. The GCM1 mRNA expression is only induced by less than two fold upon EVT and ST differentiation [$\text{Log}_2(\text{FPKM}+1)=4.5, 7.2$ and 6.5 respectively in TSCs, EVTs and STs]. It is not clear how authors will explain this. They should do a through comparison of their cells with respect to the Okae cells.

10. The derivation of human TSC from term placenta needs better characterization. For example, it is important to show (i) how global gene expression level during the derivation process, (ii) What happens if GCM1 is ectopically expressed or ΔNP63 is depleted during the derivation process (not after derivation) (iii) How does hypoxia promotes to establish stem ness, (iv) what happens if term TSCs are cultured without hypoxia, (v) what is the genomic integrity (chromosomal composition) of derived term TSCs upon culturing, (vi) A detailed comparison of actual gene expression levels in stem vs. differentiated states with respect to the TSCs derived from first-trimester placenta.

11. The in vivo transplantation data is not convincing. There are almost no CK7 or HLA-G positive cells in panel b of Figure S2. Also GCM1-KO cells have very low amount of CK7 positive cells, which is surprising as loss of GCM1 should promote proliferation. The in vivo analyses should be a main figure, and need to be better analyzed with injections in multiple mice and quantitative data.

Response to the comments of Reviewer #1

The authors describe a mechanism by which trophoblast stem (TS) cells may be maintained in the human placenta. Specifically, they claim that p63 (specifically the deltaNp63alpha isoform) maintains stemness and GCM1 promotes differentiation and the functional relationship between the two proteins determines the balance between these two states. Human TSC derivation has previously been achieved using culture additives, which in this manuscript are termed EGF/CASVY; however, the mechanism by which this culture condition maintains TSC has not been determined. The authors of this manuscript claim that EGF/CASVY activates deltaNp63alpha, thus suppressing GCM1 activity, and promoting TS “conversion” from BeWo choriocarcinoma cells as well as cytotrophoblast (CTB) derived from term placenta. However, term CTB-derived TSC using EGF/CASVY is only stably achieved under hypoxic conditions (1% oxygen) where deltaNp63alpha is further upregulated and GCM1 activity is further downregulated.

Further, the authors also point to CKMT1, a GCM1 target, as crucial for syncytiotrophoblast (STB) differentiation and a protein whose expression is decreased in patients with preeclampsia.

Overall there is a significant amount of data to support a ying-yang relationship between p63 and GCM1. However, the signaling relationships are more complicated than claimed in the paper. Furthermore, the authors do not in fact establish that the mechanism of EGF/CASVY media can be attributed to the p63-GCM1 relationship, nor do they establish that the ability to derive TSC from term CTB (which has failed in the past) is in fact due to suppression of GCM1 expression by hypoxia. In addition, the in vivo tumor formation assay data as well as RNAseq data showing that term CTB have in fact been converted to TSC are at best weak. Nevertheless, the protocol for derivation of TSC from term CTB would be a valuable tool for the field. Further, the relationship between GCM1-CKMT1 and STB differentiation and preeclampsia is also novel and significant. Specific points (both major and minor) are listed below, which, if addressed, would help align the conclusions with the results.

Major points:

- 1) The phrase “TS conversion” is a bit problematic, particularly in context of the application of EGF-CASVY media to blastocyst and first trimester placental tissue for derivation of TSC. It makes more sense when applied to pluripotent stem cells (that these cells are being “converted” to TSC); but in context of

tissues, the media is likely not “converting” one cell type into another, but rather capturing a subpopulation of CTB in a particular stem-like state.

→ We thank the Reviewer for this constructive comment. We tried to use “TS cell conversion” to cover the derivation of TS^{Term}, TS^{CT}, TS^{blast}, TS^{naive}, and iTS cells from different cell types by EGF/CASVY. In the present study, we demonstrated that ITGA6⁺ term CTBs contain a heterogeneous population of GCM1⁺, ΔNp63α⁺, and GCM1⁺/ΔNp63α⁺ CTBs (Fig. 3b, upper panel), which could be converted into a homogeneous population of ΔNp63α⁺ CTBs by EGF/CASVY (Fig. 3b, lower panel). In addition, multidimensional scaling analysis supported the conversion of ITGA6⁺ term CTBs into TS^{Term} cells (Fig. 6a). We believe that EGF/CASVY modulates the ΔNp63α-GCM1 antagonism to facilitate establishment of the above-mentioned TS cell types. Currently, we are investigating the mechanism by which EGF/CAVY maintains TS cells.

2) While the physical interaction and reciprocal expression between p63 and GCM1 seems sound, the relationship is only characterized for GCM1 regulation of p63. The authors do acknowledge the in the text, but the title implies two-way regulation.

→ In the present study, we showed that GCM1 interacts with ΔNp63α and promotes ΔNp63α destabilization. To examine how ΔNp63α downregulates GCM1 activity, we demonstrated that ΔNp63α does not directly inhibit GCM1 activity. Instead, ΔNp63α indirectly inhibits GCM1 activity through GATA3, which is known to inhibit the transcriptional activity of GCM1. Our study indicates that direct and indirect mechanisms are involved in the reciprocal and antagonistic regulation between ΔNp63α and GCM1.

3) The authors claim that p63 inhibits GCM1 function and not GCM1 expression. This conclusion was likely reached due to the binding at the transactivation domain and the unchanged expression of GCM1 upon manipulation of p63 expression. However, the readout of GCM1 function/activity was the activation of differentiation genes. To be able to make this conclusion, the authors would need to show reduced GCM1 binding by ChIPseq in cell lines overexpressing p63. Alternatively, increasing amounts of GCM1 in the presence of overexpressed p63 should “rescue” expression of HTRA4 and hCGbeta.

→ We thank the Reviewer for this constructive comment. We have introduced HA-tagged GCM1 into the BeWo cells stably expressing ΔNp63α-FLAG and

demonstrated that expression of HTRA4 and hCG β can be rescued in the presence of exogenous GCM1 (Supplementary Fig. 1e).

4) The authors conclude “..EGF/CASVY induces dedifferentiation of BeWo cells by activation of Δ Np63 α and trophoblast stemness gene expression, and thereby suppress GCM1 activity and differentiation-related target gene expression.” The authors do show EGF/CASVY induces dedifferentiation and increases p63 and trophoblast stemness genes. However, this can only be concluded if EGF/CASVY does not induce stemness genes in the absence of p63.

→ Because EOMES is unlikely to play a role in the regulation of human trophoblast stemness, we now focused on the trophoblast stemness genes *ELF5*, *TEAD4*, and *EPCAM* (please also see our answer to comment #9). We compared expression of *ELF5*, *TEAD4*, and *EPCAM* in the EGF/CASVY-treated scramble control and Δ Np63 α -knockdown BeWo cells by qRT-PCR analysis. Indeed, Δ Np63 α knockdown impeded the induction of *ELF5*, *TEAD4*, and *EPCAM* by EGF/CASVY in BeWo cells (Supplementary Fig. 3). We believe that induction of trophoblast stemness gene expression by EGF/CASVY is very likely associated with Δ Np63 α . In addition, the expression of *ELF5*, *TEAD4*, and *EPCAM* was downregulated by Δ Np63 α knockdown in JEG3 cells (Supplementary Fig. 1f). These observations suggested a gene regulatory network of Δ Np63 α and trophoblast stemness genes in the regulation of trophoblast stemness. We have described this possibility in the Discussion section.

5) Similarly, hypoxia is concluded to downregulate GCM1 promoter activity (and therefore GCM1 expression). However, the details of hypoxia treatment are neither described, nor confirmed (i.e. by evaluation of HIF components). Also, hypoxia cannot necessarily be deemed causative here, unless GCM1 overexpression in hypoxic conditions reverses the increase in p63 and decrease in hCG β .

→ In the Methods section of the present study, we mentioned that hypoxia was achieved by exposing cells to 1% O₂, 5% CO₂, and 94% N₂ in a multigas incubator (Astec, Fukuoka, Japan), whereas normoxia was achieved with 21% O₂, 5% CO₂, and balanced N₂. To confirm the hypoxia treatment, we have measured HIF1 α protein levels in normoxic and hypoxic TS^{Term}#2 cells. As shown in Fig. R1, induction of HIF1 α was observed in the TS^{Term}#2 cells under hypoxia, but not normoxia.

Fig. R1 TS^{Term}#2 cells were incubated under normoxic (N) or hypoxic (H) conditions for 72 h. Cells were harvested for coimmunoprecipitation analysis using HIF1α antibody. Total lysates (input) were subjected to immunoblotting analysis using β-actin antibody.

In Fig. 3c, we demonstrated that EGF/CASVY enhances ΔNp63α expression and suppresses GCM1 expression by immunofluorescence microscopy, which was reversed after withdrawal of EGF/CASVY. This observation was also confirmed by quantitative RT-PCR (Fig. R2). Importantly, the effects of EGF/CASVY on ΔNp63α and GCM1 expression were further enhanced by hypoxia (Fig. 3d). These results strongly supported hypoxia is an important regulator of GCM1 downregulation in the derivation of TS^{Term} cells from term CTBs. Because hypoxia induces GCM1 degradation (*JBC*, 284: 17411, 2009), we think overexpression of GCM1 in hypoxic conditions may complicate data interpretation.

Fig. R2 Regulation of ΔNp63α and GCM1 expression by EGF/CASVY. ITGA6⁺ term CTBs were incubated with or without EGF/CASVY for 7 days. Cells were harvested for quantitative RT-PCR of ΔNp63α, GCM1, and hCGβ transcripts. Mean values and the standard deviation obtained from three independent experiments are presented.

6) Many of the experiments are done using choriocarcinoma cells, which are known to be suboptimal models of in vivo human trophoblast. For this reason, at the very least, all figures should be properly labeled to indicate the cell type being used. It might best, however, to repeat at least some of these experiments in TS-term cells.

→ We have labelled the cell types being used in all figures. To investigate the antagonism between ΔNp63α and GCM1 in TS^{Term} cells, we have tested the effect of ΔNp63α overexpression on the differentiation of TS^{Term} cells into STBs. We

demonstrated that $\Delta Np63\alpha$ inhibits STB differentiation by suppressing GCM1-mediated hCG β expression and cell fusion in the $\Delta Np63\alpha$ -expressing ST-TS^{Term} cells (Fig. 4g). In addition, we have shown that GCM1 knockout impairs the differentiation of TS^{Term} cells into STBs and EVTs (Supplementary Fig. 5).

7) lines 74-76: The meaning and relevance of this sentence are unclear: JEG3 are choriocarcinoma cells, with abnormal ploidy and inability to make syncytiotrophoblast. Their ability to proliferate and invade is a property of their origin from a tumor; that they express CDX2, ELF5, and C19MC is because they originate from a choriocarcinoma. To say that they may be better than BMP4-treated hESC as a model for bona fide human TS because of expression of a handful of markers is disingenuous at best.

→ We thank the Reviewer for this comment. We do not have evidence to support that JEG3 cells are better than BMP4-treated hESCs in terms of the characteristics of human TS cells. We have corrected the sentence “In this regard, the JEG3 choriocarcinoma cell line expressing CDX2, ELF5, and *C19MC* exhibits better characteristics of TS cells “ into “In this regard, the JEG3 choriocarcinoma cell line expressing CDX2, ELF5, and *C19MC* exhibits some characteristics of TS cells” in the revised manuscript.

8) Lines 113-116: It is misleading to state that the role of p63 in TS maintenance has not been previously addressed (see References #26 and 27, as well as Lee et al. Human Pathology, 2007).

→ We apologize for this misleading. We have provided the following information to address the role of $\Delta Np63\alpha$ in the maintenance of TS cells: “p63 is a family member of the p53 tumor suppressor. $\Delta Np63\alpha$, an isoform of p63 transcription factor produced by alternative promoter usage and splicing, is highly expressed in TS cell-like CTBs. Importantly, $\Delta Np63\alpha$ levels in CTBs are decreased during STB differentiation and $\Delta Np63\alpha$ overexpression inhibits hCG expression in differentiating CTBs and cell migration of JEG3 cells, implying a role for $\Delta Np63\alpha$ in regulation of STB and EVT differentiation. $\Delta Np63\alpha$ is also predominantly expressed in the stem cells of stratified epithelia. These observations suggested a possible role of $\Delta Np63\alpha$ in TS cell maintenance.” We have also cited the *Hum Pathol* paper by Lee *et al.*

9) A lot of extrapolation is made from mouse TSC to human TSC, despite extensive data supporting significant differences between these two species. Importantly, EOMES has not been identified as a TSC-associated gene in the human placenta, and has in fact been found to be absent in both human

blastocyst (Blakeley et al. 2015) and early gestation trophoblast (Soncin et al. 2018). It should be kept in mind that JEG3 and BeWo cells are choriocarcinoma cells and as such likely express many genes NOT found in primary human TSC.

→ We agree with the Reviewer that there is significant difference between mouse and human TS cells. EOMES expression was not reported in the RNA-seq datasets of TS^{blast} and TS^{CT} cells, nor was detected in TS^{Term} cells in our RNA-seq analyses. Accordingly, we have removed the EOMES data in the revised manuscript and concentrated on the trophoblast stemness genes *ELF5*, *TEAD4*, and *EPCAM*, which are all expressed in TS^{CT}, TS^{blast}, and TS^{Term} cells.

10) Figure 1h: Not sure I see the “effect of FSK on Δ Np63 α expression was abrogated by GCM1 knockdown” since (at least by eye) the downregulation of p63 following FSK treatment is similar with either scramble or GCM1 shRNA. It might help if the authors showed a quantification of the western blot (i.e. underneath the blot). However, it also appears that at the RNA level, there isn't any significant difference in p63 expression under these different conditions.

→ We thank the Reviewer for this comment. We have performed densitometry to quantitate the immunoblot bands of GCM1 and Δ Np63 α in the left panel of revised Fig. 1h. In addition, we have replotted the qRT-PCR data for better presentation of the GCM1 and Δ Np63 α mRNA levels in the right panel of revised Fig. 1h.

11) Figure 2b: If the authors are trying to implicate a role for p63 in indirect inhibition of GCM1 activity, they would need to knockdown GCM1 in the context of p63 knockdown and show that the HTRA4 promoter activity being measured here is in fact GCM1-dependent.

→ HTRA4 is primarily expressed in human placentas. Regulation of HTRA4 promoter by GCM1 and downregulation of GCM1 activity by GATA3 have been reported in our previous studies (*Mol Cell Biol*, 32: 3707, 2012; *Sci Rep* 6: 21630, 2016). In the present study, we demonstrated that Δ Np63 α regulates GATA3 expression because Δ Np63 α overexpression in BeWo cells increases GATA3 expression, whereas Δ Np63 α knockdown in JEG3 cells decreases GATA3 expression (Fig. 2a). We now further demonstrated that Δ Np63 α stimulates GATA3 promoter activity using a GATA3 promoter reporter construct, pGATA3-0.5Kb (Supplementary Fig. 2 and please also see our answer to comment #7 of Reviewer #2). These results suggested that Δ Np63 α may directly upregulates GATA3 to suppress GCM1 activity. We thank the Reviewer for the suggested experiment. We would like to pointed out that GCM1 knockout abolishes HTRA4 expression in JEG3 cells given that Δ Np63 α

expression is not suppressed by FSK (Fig. 2f). In addition, suppression of HTRA4 expression in the Δ Np63 α -FLAG-expressing BeWo cells was reversed by GCM1 overexpression (Supplementary Fig. 1e). Collectively, our data strongly suggested that Δ Np63 α suppresses GCM1 target gene expression through GATA3-mediated inhibition of GCM1 activity.

12) Continuous passage under hypoxia (1% oxygen) is a very stringent oxygen tension for cellular maintenance and in fact has been shown to induce significant metabolic stress, even in trophoblast. This needs to at least be discussed, as a potential issue for longterm maintenance of TS-term cells. In addition, the actual method for maintaining hypoxic culture is not explained in detail: what type of incubator was used, and were cells maintained under these conditions even during feeding and passage? In Figure 4e, when TS-term are shown under “normoxia,” were these cells temporarily exposed to higher oxygen or actually passaged under normoxia?

→ The hypoxic culture conditions in the present study were achieved by exposing cells to 1% O₂, 5% CO₂, and 94% N₂ in a multigas incubator (Astec, Fukuoka, Japan). The normoxic culture conditions were achieved with 21% O₂, 5% CO₂, and balanced N₂. TS^{Term} cells were maintained under hypoxia and transferred to normoxia for differentiation assays. For feeding and passage, TS^{Term} cells were temporarily exposed to the ambient temperature and oxygen in the laminar flow hood. For the experiments in Fig. 4e, TS^{Term} cells were incubated and induced for STB differentiation under normoxia for 72 h before harvesting for analysis. As a regular practice in cell culture, we conducted experiments in choriocarcinoma cells and TS^{Term} cells that have been maintained for no more than ten passages. An ongoing study in our laboratory is to improve the culture conditions with additional small-molecule inhibitors to efficiently suppress GCM1 expression in term CTBs in order to establish TS^{Term} cells under normoxia. This may help to prevent the metabolic stress induced by hypoxia.

13) The authors emphasize reciprocal expression of p63 and GCM1, but Figure 4g in fact does not show suppression of GCM1 expression following forced expression of p63 in TS-term cells, even following FSK treatment. How can this be explained?

→ In the present study, the levels of GCM1 transcript and protein were not significantly changed in the BeWo (Fig. 1e) and TS^{Term} (Fig. 4g) cells overexpressing Δ Np63 α . Although Δ Np63 α directly interacts with GCM1, Δ Np63 α indirectly suppresses GCM1 activity through GATA3 to downregulate GCM1 target gene

expression. We believe that reciprocal regulation of $\Delta Np63\alpha$ and GCM1 activities plays a crucial role in trophoblast stemness and differentiation.

14) Supplementary Figure 2b: It is not clear that the lone HLAG+/CK7+ cell is in fact part of the tumor formed by the TS-term cells. In the absence of such cells within the tumor, it cannot be said that these cells are in fact bipotential *in vivo*.

→ We thank the Reviewer for this constructive comment. The purposes of these experiment were to demonstrate the *in vivo* differentiation potency of TS^{Term} cells and that GCM1 is required for the differentiation of hCG β ⁺ or HLA-G⁺ cells from TS^{Term} cells *in vivo*. The original IHC picture of HLA-G⁺ cells derived from wild-type (WT) TS^{Term} cells was misleading by showing few positive cells. Using another commercial anti-HLA-G mouse mAb (Santa Cruz Cat# sc-21799), we have produced better data to show more HLA-G⁺ cells derived from WT TS^{Term} cells (Fig. 5b). Expression of hCG β and HLA-G was not detected in the lesion derived from GCM1-KO TS^{Term} cells (Fig. 5c).

15) The authors claim that they have generated cells similar to TSC derived from first trimester and blastocysts, yet the RNAseq comparison does not include any other cell type (including the cell of origin, term CTB). The authors need to show that in fact TS-term move away from being term-CTB-like, and toward a TS-CT/TS-blast-like state.

→ We have performed RNA-seq analyses in ITGA6⁺ term CTBs (#1-#3), TS^{Term} (#1-#3) cells and their derivative STBs (ST-TS^{Term}#1-#3) and EVT^s (EVT-TS^{Term}#1-#3). Multidimensional scaling of protein-coding genes in the RNA-seq datasets showed dissimilarities among ITGA6⁺ term CTB, TS^{Term}, ST-TS^{Term}, and EVT-TS^{Term} cell types (Fig. 6a). These results supported that TS^{Term} cells move away from being ITGA6⁺ CTBs. High correlation between TS^{Term} cells and TS^{CT} and TS^{blast} cells was observed by Pearson correlation analysis (Fig. 6c).

16) lines 384-386: Reference #36 shows ELF5 expression in early placenta but does not provide evidence for a role for ELF5 in “establishment” of the TS compartment. This reference also does not discuss EOMES.

→ We thank the Reviewer for this comment. As mentioned in our answer to comment #9, we have removed EOMES to avoid confusion that may arise from the differential gene expression between mouse and human TS cells. In addition, we have removed the statement about ELF5 in the revised manuscript due to the uncertain role of ELF5 in human TS cell maintenance.

17) lines 397-400: that EGF/CASVY may stimulate p63 expression is not supported by data and is therefore highly speculative.

→ We showed that EGF/CASVY treatment stimulates Δ Np63 α expression in BeWo cells (Fig. 3a). In addition, IHC analysis indicated that Δ Np63 α expression is decreased in TS^{Term} cells after withdrawal of EGF/CASVY (Fig. 3c). Although the underlying mechanism remains elusive, we wish to investigate this issue in the near future.

18) Line 444 (“lifetime storage”): Did the authors characterize the ability to freeze-thaw these TS-term cells and still maintain bipotency?

→ For each placental biopsy (~40 g), we obtained approximately 2×10^7 CTBs and froze the cells into 4 cryotubes. Two cryotubes (1×10^7 cells) were thawed for derivation of TS^{Term} cells, which were amplified and frozen at passage 4 or 5. The frozen TS^{Term} cells could be recovered after thawing and maintained bipotency in functional studies. We think it is possible to scale up preparation of ITGA6⁺ term CTBs and derivation of TS^{Term} cells from the prepared ITGA6⁺ term CTBs.

Minor points:

1) On all immunoblots for p63, it would be clearer if the arrow always pointed to the specific band.

→ We have added arrow and arrowhead symbols to point out the Δ Np63 α band and the non-specific band, respectively, on all immunoblots.

2) Is Figure 1e the same as Supplementary Figure 1d?

→ Both figures present similar observations that Δ Np63 α overexpression suppresses GCM1 target gene expression. In Fig. 1e, the BeWo cells stably expressing Δ Np63 α were generated by puromycin selection. In Supplementary Fig. 1d, the BeWo cells stably expressing Δ Np63 α and GFP were sorted by flow cytometry. We have provided information to clarify this issue in the revised manuscript.

3) Line 229 – data not shown, would be helpful if at least the assay is described for this conclusion.

→ Transient expression experiments were performed in JEG3 cells overexpressing HA-tagged ubiquitin, followed by coimmunoprecipitation analysis to measure the level of ubiquitinated Δ Np63 α . In this scenario, the level of ubiquitinated Δ Np63 α was not significantly changed in the presence or absence of FSK. We have modified the original description to “However, *in vivo* ubiquitination assay performed in JEG3

cells expressing HA-tagged ubiquitin showed that ubiquitination of $\Delta Np63\alpha$ is not affected by GCM1 or FSK (data not shown).”

4) Line 280: “suppressive effects of hypoxia on GCM1 ‘activity’” the data describes effects on GCM1 expression. The distinction between “activity” and “expression” is especially important to avoid confusion after the results in the previous section.

→ We apologize for the confusion about “GCM1 activity” and “GCM1 expression.” Hypoxia downregulates GCM1 autoregulation at the transcriptional level and also stimulates GCM1 degradation at post-translational level (*J Biol Chem*, 284: 17411, 2009). We believe that both events contribute to suppression of GCM1 activity in the hypoxic TS^{Term} cells. In the present study, we showed downregulation of GCM1 autoregulation in TS^{Term} cells by hypoxia (Fig. 4f). We have changed “suppressive effects of hypoxia on GCM1 activity” into “suppressive effects of hypoxia on GCM1 expression.”

5) Figure 2e. This blot does not seem to have the nonspecific band for $\Delta Np63\alpha$? Or is that the specific band? It would be good to consistently provide the arrow to the specific band.

→ The non-specific immunoblot band appears after a long exposure of blot to film. The original picture was from an experiment with a short exposure. We have repeated the experiment with a long exposure of blot to film in Fig. 2e. The $\Delta Np63\alpha$ and the non-specific bands were marked by the arrow and the arrowhead, respectively.

6) Figure 2f. Same as figure 2e, the arrow and molecular weights are missing for p63.

→ The experiment was repeated and images of short and long exposures for $\Delta Np63\alpha$ were provided in the revised Fig. 2f. In addition, molecular weight, arrow, and arrowhead markers were added in the revised figure as in Fig. 2e.

7) Antibody catalogue #s and dilutions or concentrations should be provided.

→ The catalogue numbers of antibodies and the dilutions used in the present study have been provided in Reporting Summary.

Response to the comments of Reviewer #2

The manuscript by Wang et al. described an interesting mechanism between GCM1 and Δ NP63 that dictates self-renewing stem state vs. differentiation of human trophoblast stem cells (Human TSC). Authors also used this mechanistic platform to develop a strategy to successfully derive Human TSCs from term placenta, which is still elusive. The biggest strength of this paper that could significantly advance the field is the successful establishment of human TSCs from term placenta. Thus, this Reviewer was very interested with the manuscript. However, the manuscript has numerous problems, contradictory data with the published papers, and most importantly insufficient data to define the core hypothesis that GCM1 and Δ NP63 antagonism is the dictating factor for TSC stemness vs. differentiation. My major concerns are mentioned below.

1. I have a major concern about the data, which is shown in Fig. 1a about GCM1 expression in a first-trimester placenta. In contrary to data shown in Fig.1a, earlier publication (Chiu and Chen, *Sci. Rep* 2016 Feb 22;6:21630. doi: 10.1038/srep21630.) reported that GCM1 is expressed in undifferentiated cytotrophoblasts (CTBs, both villous and column CTBs). A quick look of our own single cell RNA-seq data in human first-trimester placentae shows that GCM1 mRNA is expressed in undifferentiated CTBs, although the expression is induced during differentiation. Also, the supplementary Fig. S1a clearly shows GCM1 expression in undifferentiated CTBs. Thus, the data about GCM1 expression, shown in Fig. 1a, does not match with the other observations, which is a major concern. If GCM1 is expressed in undifferentiated CTBs, the whole hypothesis that suppression of GCM1 is important for self-renewing stem state needs to be revisited. Both immunostaining and in-situ hybridization data with multiple placental sections showing both floating and anchoring villi should be included.

→ We thank the Reviewer for this constructive comment. We have reported GCM1 expression in the first-trimester CTBs by IHC analysis in our previous studies (*Sci Rep*, 6: 21630, 2016; *FASEB J*, 27: 2818, 2013). In these studies, we mentioned that GCM1 is detected in CTBs, but did not conclude that GCM1 is expressed in the “undifferentiated” CTBs. Given the rapid growth and development of placenta in the first trimester, it is very possible that differentiation of CTBs into STBs and EVT_s is highly dynamic in different placental villi. Indeed, the present study may provide an explanation to the observation of GCM1 expression in CTBs. We believe that

first-trimester villi harbor a population of undifferentiated and differentiating CTBs. The GCM1-expressing CTBs are very likely the CTBs being induced to differentiate into STBs or EVT. The single-cell RNA-seq analysis of 3D-cultured human blastocysts by Xiang *et al.* (*Nature*, 577: 537, 2020) categorized trophoblast populations into six subpopulations: pre-CTBs (TEAD4⁺HLA-G⁻), post-CTBs, early STBs (CGB⁺CSH1⁻HLA-G^{low}), STBs, early EVT (HLG-A⁺CSH1⁺MMP2⁺ERBB2⁺) and EVTs on the basis of developmental time and marker genes. We performed meta-analyses of the datasets and showed that GCM1 expression is barely detectable in pre-CTBs, but significantly increased in post-CTBs and the STB and EVT subpopulations (Fig. R3). This study suggests that GCM1 is expressed in a CTB subpopulation ready for STB and EVT differentiation. Of note, we have replaced the original IHC image of first-trimester placenta with another image in order to show similar intensities of GCM1 and Δ Np63 α signals in first-trimester and term placentas in Fig. 1a. We have also performed RNAscope *in situ* hybridization to demonstrate coexpression of GCM1 and Δ Np63 α transcripts in the CTBs of first-trimester and term placentas (ISH, Fig. 1a).

Fig. R3 GCM1 expression in trophoblasts during early human embryonic development *in vitro*. Meta-analysis of GCM1 gene expression in the single-cell RNA-seq datasets (GSE136447) of 3D-cultured human blastocysts shows that GCM1 is differentially expressed between pre- and post-CTB subpopulations.

2. Why the co-immunoprecipitation experiments are performed with ectopic overexpression system in HEK 293 cells? What are the ectopic protein expression levels compared to endogenous expression levels in trophoblast cells? As authors have good antibodies, they should test it in term CTBs.

→ We believe that coexpression of Δ Np63 α and GCM1 is a transient event during the differentiation of CTB into STBs and EVTs in placenta. Because GCM1 promotes Δ Np63 α degradation, the time window of Δ Np63 α ⁺→GCM1⁺/ Δ Np63 α ⁺→GCM1⁺ transition is expected to be narrow. Indeed, the number of GCM1⁺/ Δ Np63 α ⁺ CTBs is less than 10% of the ITGA6⁺ term CTB population (Fig. 3b). We have described this

issue in the Discussion section. On the other hand, we have tried to enhance GCM1 expression in normoxic TS^{Term} cells for coimmunoprecipitation analysis. However, the number of GCM1⁺/ΔNp63α⁺ TS^{Term} cells under normoxia for 48 h (data not shown) or 72 h (Fig. R4) was few, indicating rapid GCM1 upregulation and ΔNp63α downregulation. Without a sufficient number of GCM1⁺/ΔNp63α⁺ cells, we were unable to demonstrate the interaction between endogenous GCM1 and ΔNp63α by coimmunoprecipitation analysis. Fortunately, ΔNp63α overexpression did not significantly affect the endogenous GCM1 level in BeWo cells (Fig. 1e). Accordingly, we expressed ΔNp63α-FLAG in BeWo cells and demonstrated the interaction between endogenous GCM1 and ΔNp63α-FLAG (Fig. 1d).

Fig. R4 Expression of ΔNp63α and GCM1 in normoxic TS^{Term} cells. TS^{Term}#2 cells were incubated under normoxic conditions for 72 h and then subjected to immunofluorescence microscopy analysis using ΔNp63α and GCM1 antibodies. Scale:

3. The JEG3 cell line is choriocarcinoma cell line. It is not a true representative of primary trophoblast cells. Thus molecular mechanism in that context may not be definitive for primary CTBs.

→ We agree with the Reviewer that JEG3 choriocarcinoma cells may not be a good model for primary CTBs. We have corrected the sentence “In this regard, the JEG3 choriocarcinoma cell line expressing CDX2, ELF5, and *C19MC* exhibits better characteristics of TS cells “ into “In this regard, the JEG3 choriocarcinoma cell line expressing CDX2, ELF5, and *C19MC* exhibits some characteristics of TS cells” in the revised manuscript.

4. Surprisingly, different experiments with JEG3 cells in Fig. 1 show different levels and banding patterns of same protein. For example in Fig. 1C, GCM1 protein is readily detectable in JEG3 cells without FSK treatment. However, the presented data in Fig. 1h shows almost undetectable GCM1 protein under same experimental condition. The western blot band of ΔNP63 is detected as

a single band in some experiments (Fig. 1C, 1e, 1F) and as a double band in other experiments (Fig. 1G, 1H).

→ We thank the Reviewer for this constructive comment. The inconsistent patterns of immunoblot bands of GCM1 and Δ Np63 α were due to different exposure times used in different experiments. Considering the differential expression of Δ Np63 α and GCM1 in BeWo and JEG3 cells, a longer exposure time was used to reveal the GCM1 band in JEG3 cells in Figs. 1c and 1g. Unfortunately, the commercial Δ Np63 α antibody recognized a non-specific band below the Δ Np63 α band in immunoblotting analysis after a long exposure of blot to film. After knocking down Δ Np63 α , the Δ Np63 α band disappeared, but not the non-specific band (Fig. 1h). We have further confirmed the Δ Np63 α band by coimmunoprecipitation analysis (Fig. R5).

Fig. R5 Specificity of the Δ Np63 α antibody. JEG3 cells were subjected to immunoblotting or coimmunoprecipitation analysis using Δ Np63 α antibody. Arrow and arrowhead indicate the Δ Np63 α band and a non-specific band, respectively. Short- and long-exposure images of immunoblot bands are presented. Total lysates (input) were diluted 3.3 times for coimmunoprecipitation analysis and subjected to immunoblotting analysis using β -actin antibody.

The single band recognized by the Δ Np63 α antibody in Fig. 1e was most likely the Δ Np63 α -FLAG in the BeWo cells stably expressing Δ Np63 α -FLAG. In Fig. 1f, the single band recognized by the Δ Np63 α antibody was a non-specific band in the scramble BeWo cells and an additional Δ Np63 α band was detected above the non-specific band in the GCM1-knockdown BeWo cells. We have added arrows and arrowheads to point out the Δ Np63 α and the non-specific bands on all immunoblots in the revised manuscript.

5. The GCM1 knockdown efficiency in FSK treated JEG3 cells (Fig. 1H) and corresponding change in Δ NP63 protein is minimal. Thus, the claim in lines 185 and 186 that GCM1 suppresses Δ NP63 activity based on that data is surprising and not conclusive. Also, any change in expression does not mean GCM1 regulates activity of Δ NP63.

→ We have performed densitometry analysis to quantitate the GCM1 and Δ Np63 α bands in the immunoblotting analysis of FSK-treated scramble and

GCM1-knockdown JEG3 cells in Fig. 1h. GCM1 knockdown increased the Δ Np63 α protein level by 61% and abrogated hCG β expression. With the same GCM1 knockdown strategy, we also observed a similar effect on hCG β expression in BeWo cells in a previous study (Fig. 3B, *Mol Cell Biol* 36: 197, 2016). We agree with the Reviewer that the observed change in the Δ Np63 α protein level does not mean GCM1 regulates Δ Np63 α activity. We have changed the statement in lines 185 and 186 to “The effect of FSK on Δ Np63 α expression was compromised by *GCM1* knockdown confirming the role of GCM1 in the repression of Δ Np63 α expression during STB differentiation (Fig. 1h).”

6. The increase in differentiation efficiency (Fusion index) of JEG3 cells with FSK is only ~15%. This is a very narrow window to generate any conclusive data. Authors should try induction and repression of cell differentiation upon GCM1 and Δ NP63 overexpression with actual human TSCs.

→ We have generated TS^{Term} cells expressing Δ Np63 α -FLAG and demonstrated that expression of hCG β and Syncytin-1 as well as cell fusion are significantly suppressed in the Δ Np63 α -FLAG-expressing ST-TS^{Term} cells compared with the mock ST-TS^{Term} cells (Fig. 4g). In addition, cell fusion and hCG β expression were significantly decreased in the GCM1-knockout ST-TS^{Term} cells (Supplementary Figs. 5a and 5b).

7. There is no real data showing GATA3 expression is suppressed by Δ NP63. The actual reference indicated regulation of GATA3 by the other isoform TP63 (the whole protein) in hair follicle cells. Thus, direct regulation of GATA3 by Δ NP63 is purely speculative.

→ Our results suggested that Δ Np63 α upregulates GATA3 expression. BeWo and JEG3 cells exhibit lower and higher levels of endogenous Δ Np63 α , respectively (Fig. 1c). Along this line, we showed that Δ Np63 α overexpression increases GATA3 expression in BeWo cells and Δ Np63 α knockdown decreases GATA3 expression in JEG3 cells (Fig. 2a). These results suggested that GATA3 is downstream of Δ Np63 α . We now performed transient expression experiments to demonstrate that Δ Np63 α stimulates GATA3 promoter activity. The luciferase activity directed by a GATA3 promoter reporter plasmid, pGATA3-0.5Kb, was significantly upregulated by Δ Np63 α (Fig. R6). The results are presented in Supplementary Fig. 2 of the revised manuscript. Therefore, we believe that Δ Np63 α may directly stimulate GATA3 expression to suppress GCM1 activity.

Fig. R6 Upregulation of GATA3 promoter activity by Δ Np63 α . **a**, Schematic representation of GATA3 promoter. Two candidate p63-binding sites (p63bs1, and-2) are listed. Arrows indicates the transcriptional start site. **b**, Δ Np63 α stimulates GATA3 promoter activity directed by the reporter plasmid, pGATA3-0.5Kb. Mean values and the standard deviation obtained from three independent experiments are presented.

GATA3 is listed as a p63 target genes in Fig. 1 of reference #29 (Vigano MA and Mantovani R. Hitting the numbers: the emerging network of p63 targets. *Cell Cycle*, 6: 233, 200) based on the study by Candi *et al.* (p63 is upstream of IKK alpha in epidermal development. *J Cell Sci*, 119: 4617, 2006). In Fig. 4A of Candi's paper, Δ Np63 α was shown to activate GATA3 promoter. We have also cited this paper in the revised manuscript.

8. Also, according to the manuscript GATA3 is a negative regulator of GCM1. However, GATA3 and GCM1 both are abundantly expressed and functions in primary differentiated STs of actual human placenta. Thus, the conclusion drawn from studies in choriocarcinoma cell lines does not fit with actual placenta.

→ We thank the Reviewer for this constructive comment. According to the human protein atlas database (<https://www.proteinatlas.org/ENSG00000107485-GATA3/tissue>), GATA3 is expressed in a variety of human tissues. Our previous study has revealed one of the biological functions of GATA3 in placenta, i.e. GATA3 interacts with GCM1 and inhibits its transcriptional activity (*Sci Rep*, 6: 21630, 2016). Although IHC indicated that GATA3 and GCM1 both are expressed in trophoblasts, the physiological outcomes of this observation might be context-dependent during trophoblast differentiation. Because GCM1 activity is regulated by post-translational modifications, the functional outcomes of the interaction between GCM1 and GATA3 may be regulated by different signaling pathways during trophoblast differentiation. We wish to study this intriguing question in the near future.

9. As I mentioned earlier, the major strength of this manuscript is the

successful derivation of Human TSC lines from term placenta. However, the GCM1 antagonism mechanism, which is indicated in this paper, is surprising. Beacuses, the human TSC lines derived from first-trimester human placenta and reported by (Okae et. al., 2018, Cell Stem Cell) expresses significant amount of GCM1 mRNA. The GCM1 mRNA expression is only induced by less than two fold upon EVT and ST differentiation [(Log₂(FPKM+1)=4.5, 7.2 and 6.5 respectively in TSCs, EVTs and STs]. It is not clear how authors will explain this. They should do a through comparison of their cells with respect to the Okae cells.

→ We agree with the Reviewer that GCM1 is expressed in TS^{blast} and TS^{CT} cells based on their RNA-seq datasets. One possibility is that EGF/CASVY is not able to completely suppress GCM1 expression in TS^{blast} and TS^{CT} cells. Recently, Cinkornpumin *et al.* used Okae's TS cells (hTSCs: TS^{CT1}, TS^{CT3}, and TS^{BLAST2} cells) to study trophoblast differentiation (*Stem Cell Rep*, 15: 198, 2020). In the Experimental Procedures section of Cinkornpumin's paper, the authors mentioned: "We observed that reduced oxygen levels promote hTSC self-renewal but inhibit directed differentiation, so we cultured hTSCs in 5% O₂ 5% CO₂ but performed differentiation to EVT or STB at 20% O₂ 5% CO₂." We believe that the hypoxic conditions decrease GCM1 expression in Okae's TS cells to prevent differentiation and enhance cell proliferation. After inversion of the Log₂ values 4.5, 7.2, and 6.5, the GCM1 transcript levels in FPKM are 21.6, 146, and 89.5, respectively. The fold inductions of GCM1 transcript in STBs and EVTs relative to TS^{CT} and TS^{blast} cells are 4.1 and 6.8, respectively. In Fig. 7b, we compared the GCM1 transcript level in the TS^{Term} cells and ST-TS^{Term} cells under normoxia for 72 h. A 5.8-fold increase in GCM1 transcript level was observed in the ST-TS^{Term} cells. We believe that the fold induction will be much higher when compared with hypoxic TS^{Term} cells. As mentioned above, scientists have experienced difficulty in maintaining TS^{CT} and TS^{blast} cells under normoxia. We think suppression of GCM1 expression is crucial for trophoblast stemness.

10. The derivation of human TSC from term placenta needs better characterization. For example, it is important to show (i) how global gene expression level during the derivation process, (ii) What happens if GCM1 is ectopically expressed or ΔNP63 is depleted during the derivation process (not after derivation) (iii) How does hypoxia promotes to establish stem ness, (iv) what happens if term TSCs are cultured without hypoxia, (v) what is the genomic integrity (chromosomal composition) of derived term TSCs upon culturing, (vi) A detailed comparison of actual gene expression levels in stem

vs. differentiated states with respect to the TSCs derived from first-trimester placenta.

→ We thank the Reviewer for the constructive comments.

- (i) We have performed RNA-seq analyses in ITGA6⁺ term CTBs, TS^{Term} cells, and TS^{Term} cell-derived STBs and EVT. Multidimensional scaling analysis revealed dissimilarities among ITGA6-positive term CTB, TS^{Term}, ST-TS^{Term}, and EVT-TS^{Term} cell types (Fig. 6a).
- (ii) We have expressed exogenous HA-tagged GCM1 in ITGA6⁺ term CTBs, but failed to maintain the cells in EGF/CASVY under hypoxia (Fig. R7). This could be attributed to the antagonistic effect of GCM1 on Δ Np63 α that prevented cell proliferation.

Fig. R7 Derivation of TS^{Term} cells from ITGA6⁺ term CTBs is blocked by GCM1 overexpression. ITGA6⁺ term CTBs were transduced with lentiviruses harboring pCDH-GFP (mock, **a-c**) or pCDH-GFP-GCM1-HA (**d** and **e**) and incubated in EGF/CASVY under hypoxia. Cells were subcultured at day 6 (P1) and 12 (P2) and examined under a microscope. Bright-field and fluorescence images are presented. Note that GCM1-HA-expressing CTBs failed to survive after P1, whereas the mock (GFP only) counterparts could be propagated and maintained after P2. Scale: 50 μ m.

- (iii) Our results suggested that hypoxia suppresses GCM1 expression, which prevents Δ Np63 α from degradation and maintains Δ Np63 α activity in TS cells. We speculated that a gene regulatory network between Δ Np63 α and trophoblast stemness genes may be involved in regulation of trophoblast

stemness.

- (iv) Suppression of GCM1 expression by hypoxia was relieved when TS^{Term} cells in EGF/CASVY were shifted to normoxia. The normoxic TS^{Term} cells in EGF/CASVY exhibited low GCM1 levels. Upon stimulation with FSK, GCM1 expression was further enhanced in the normoxic ST-TS^{Term} cells (Figs. 4e and 7b and Supplementary Fig. 5a).
- (v) We have performed chromosomal microarray and karyotype analysis in two TS^{Term} lines (#1 and #2). Chromosomal microarray analysis using the Affymetrix CytoScan 750K arrays showed no chromosomal copy number variations in TS^{Term}#1 (P7) and TS^{Term}#2 (P5). A normal karyotype was found in TS^{Term}#1 (P10) and TS^{Term}#2 (P10). We derived TS^{Term} cells from frozen ITGA6⁺ term CTBs and the established TS^{Term} cells were amplified and frozen at early passages or subjected to analysis of GCM1 and Δ Np63 α expression and differentiation assays (please see our answer to comment #18 of Reviewer #1). As a regular practice in cell culture, we conducted experiments in choriocarcinoma cells and TS^{Term} cells that have been maintained for no more than ten passages.
- (vi) We have performed detailed comparison of the FPKM values of selected TS-, STB-, and EVT-specific genes in TS^{Term}, TS^{CT}, and TS^{blast} cells and their derivative STBs and EVTs. Because only two TS^{CT} cell lines and two TS^{blast} cell lines were examined in Okae's study, their datasets were combined for comparison with three TS^{Term} cell lines. Expression patterns of the selected lineage marker genes are comparable between TS^{Term}, TS^{TC}, and TS^{blast} cells (Fig. 6e).

11. The in vivo transplantation data is not convincing. There are almost no CK7 or HLA-G positive cells in panel b of Figure S2. Also GCM1-KO cells have very low amount of CK7 positive cells, which is surprising as loss of GCM1 should promote proliferation. The in vivo analyses should be a main figure, and need to be better analyzed with injections in multiple mice and quantitative data.

→ We thank the Reviewer for this constructive comment. We inoculated wild-type (WT) and GCM1-KO TS^{Term} cells into five NOD-SCID mice, respectively. Three lesions derived from each TS^{Term} cell types were subjected to IHC analysis. In the lesions derived from WT TS^{Term} cells, hCG β ⁺ cells were readily detectable (Fig. 5a). The original IHC picture of HLA-G⁺ cells derived from WT TS^{Term} cells was misleading by showing few positive cells. Using another commercial anti-HLA-G mouse mAb (Santa Cruz Cat# sc-21799), we have produced better data to show more HLA-G⁺ cells in the WT TS^{Term}-derived lesions (Fig. 5b). We have repeated CK7

staining in the lesions derived from GCM1-KO TS^{term} cells and presented images of CK7⁺ cells at lower and higher magnifications in Fig. 5c. Expression of hCG β and HLA-G was not detected in the lesions derived from GCM1-KO TS^{term} cells (Fig. 5c). The purposes of these experiment were to demonstrate the *in vivo* differentiation potency of TS^{term} cells and that GCM1 is required for differentiation of hCG β ⁺ or HLA-G⁺ cells from TS^{term} cells *in vivo*. We have presented the *in vivo* transplantation data in a main figure (Fig. 5).

Reviewers' Comments:

Reviewer #1:

Remarks to the Author:

The authors have addressed the majority of reviewers' concerns. Some minor points remain:

- 1) Figure 1h: can densitometry be performed on all bands on the western blot?
- 2) There are still several places in the main manuscript text, where the authors do not mention the cell type for the experiment (e.g. page 11, sentence describing Supplementary Figure 3).
- 3) Still not sure "TS conversion" is the best terminology here (this is often used when you are turning one cell type into another: pluripotent into trophoblast, fibroblast into neuron); I would use "induction of a trophoblast stem cell-like state."
- 4) Can the RNAseq data from Okae's TSC (and EVT and STB derivatives) as well as their primary CTB, EVT, and STB be analyzed together with data from term CTB-derived TSC (in Figure 6a)?
- 5) Can the authors discuss more the differences between their TS-term and Okae's TS-CT and TS-blast? Some differentially expressed genes are briefly discussed in reference to Figure 6D (bottom of page 14), but this is not discussed further.

Reviewer #2:

Remarks to the Author:

The revised manuscript addresses many of concerns and is a much improved one. However, unfortunately, a few aspects remain unclear. Following recommendations should be included to improve the manuscript. And convince the scientific community

1. I am still very concerned about using the choriocarcinoma cell lines for most of the experiments related to the interrelationship of GCM1 and deltaNP63 interaction and functions. The choriocarcinoma cell lines are not true representative of human TSCs or proliferating CTB populations. To better convince the fundamental mechanistic aspect, which is proposed in this manuscript, authors should use true human TSC lines established by Okae to test at least the most basic aspect of this study. In the rebuttal letter authors indicated that there are GCM1-high and low CTB populations. Also, the new in situ hybridization data shows GCM1 expression in CTBs of first-trimester placenta. I believe such populations can also be identified in Okae TSC lines. Authors should perform the loss of function analyses with deltaNP63 in Okae TSCs to show the effect on GCM1 expression and effect on self-renewal. Authors should also perform loss of GCM1 function analyses in those cells to show effect on deltaNP63 function (including GATA3 regulation) and the differentiation potential. Lastly, the Okae TSCs express high levels of both GCM1 and deltaNP63. So, those cells should be used to show their physical interaction.

2. GATA3 is one of the earliest genes induced during trophoblast differentiation. The data and argument related to GATA3 regulation by deltaNP63 and GATA3-mediated negative regulation of GCM1 need further validations. The new transient transfection analyses with GATA3 minimal promoter is not a convincing experiment as normally the (-)500bp region enhancer is not an optimum regulator for GATA3 gene transcription. Thus, the data is a bit surprising. Authors should perform a chromatin immunoprecipitation analyses with Okae TS lines (where deltaNP63, GATA3 and GCM1 are all expressed) to show that deltaNP63 is binding to those putative binding motifs at the GATA3 locus. In the same TSC context, authors should show that loss of GATA3 is affecting GCM1 function (I am sorry, the earlier Scientific Reports paper includes experiments performed in HTR8 cell line, which does not represent trophoblast stem cell and the argument in the rebuttal letter about this aspect is not a convincing one).

Additional Comments:

1. Introduction, line 66: References regarding Hippo signaling are selective and does not include manuscripts related to human TSCs . Authors should include following references.

Home P et al., Proc Natl Acad Sci U S A. 2012 109(19):7362-7

Meinhardt G et al., Proc Natl Acad Sci U S A. 2020 117(24):13562-13570

Saha B et al., Proc Natl Acad Sci U S A. 2020 117(30):17864-17875.

2. Introduction line 69-74, this is also a partial and incomplete information. There are several recent manuscripts, which showed that the Naïve human ES cells can be reliably converted to the trophoblast stem cell fate using the Okae EGF/CASVY medium. Authors should include this aspect and associated references in the manuscript.

Response to the comments of Reviewer #1

Reviewer #1 (Remarks to the Author):

The authors have addressed the majority of reviewers' concerns. Some minor points remain:

1) Figure 1h: can densitometry be performed on all bands on the western blot?
→ We thank the Reviewer for this comment. We have quantitated all bands on the western blot in Fig. 1h by densitometry.

2) There are still several places in the main manuscript text, where the authors do not mention the cell type for the experiment (e.g. page 11, sentence describing Supplementary Figure 3).

→ We apologize for any inconvenience caused. In the revised manuscript, we have added the information about the cell type used in the experiments in this study.

3) Still not sure "TS conversion" is the best terminology here (this is often used when you are turning one cell type into another: pluripotent into trophoblast, fibroblast into neuron); I would use "induction of a trophoblast stem cell-like state."

→ We thank the Reviewer for the valuable suggestion. We have used "induction of a trophoblast stem cell-like state" to replace "TS conversion" in the revised manuscript.

4) Can the RNAseq data from Okae' s TSC (and EVT and STB derivatives) as well as their primary CTB, EVT, and STB be analyzed together with data from term CTB-derived TSC (in Figure 6a)?

→ In Okae's study, the authors did not provide PCA data. We have incorporated the RNA-seq datasets in Okae's study and our datasets into a PCA. Over 18000 expressed genes were analyzed and our results showed dissimilarities among different placental cell types, which could be categorized in four main clusters: (1) first-trimester and term CTBs, (2) TS^{CT} , TS^{blast} , and TS^{Term} cells, (3) first-trimester STBs and the STBs derived from TS^{CT} , TS^{blast} , and TS^{Term} cells, and (4) first-trimester EVTs and the EVTs derived from TS^{CT} , TS^{blast} , and TS^{Term} cells (Fig. 6a).

5) Can the authors discuss more the differences between their TS-term and Okae' s TS-CT and TS-blast? Some differentially expressed genes are briefly discussed in reference to Figure 6D (bottom of page 14), but this is not discussed further.

→ We have provided information about the differences between TS^{CT}, TS^{blast}, and TS^{Term} cells in the Discussion section of the revised manuscript.

Response to the comments of Reviewer #2

Reviewer #2 (Remarks to the Author):

The revised manuscript addresses many of concerns and is a much improved one. However, unfortunately, a few aspects remain unclear. Following recommendations should be included to improve the manuscript. And convince the scientific community

1. I am still very concerned about using the choriocarcinoma cell lines for most of the experiments related to the interrelationship of GCM1 and deltaNP63 interaction and functions. The choriocarcinoma cell lines are not true representative of human TSCs or proliferating CTB populations. To better convince the fundamental mechanistic aspect, which is proposed in this manuscript, authors should use true human TSC lines established by Okae to test at least the most basic aspect of this study. In the rebuttal letter authors indicated that there are GCM1-high and low CTB populations. Also, the new in situ hybridization data shows GCM1 expression in CTBs of first-trimester placenta. I believe such populations can also be identified in Okae TSC lines. Authors should perform the loss of function analyses with deltaNP63 in Okae TSCs to show the effect on GCM1 expression and effect on self-renewal. Authors should also perform loss of GCM1 function analyses in those cells to show effect on deltaNP63 function (including GATA3 regulation) and the differentiation potential. Lastly, the Okae TSCs express high levels of both GCM1 and deltaNP63. So, those cells should be used to show their physical interaction.

→ We thank the Reviewer for this constructive comment. We would like to describe the logical flow of the present study. We initiated this study in 2016 to investigate the functional and physical interaction between GCM1 and $\Delta Np63\alpha$ in BeWo, JAR, and JEG3 cells. When Okae et al. reported their study in 2018, we were curious about the expression of GCM1 in TS^{CT} and TS^{blast} cells (according to Okae's RNA-seq datasets) because GCM1 regulates trophoblast differentiation and Baczyk et al. have shown that GCM1 inhibits trophoblast proliferation (*Cell Death Differ* 16: 719-727, 2009). Based upon our study of GCM1 and $\Delta Np63\alpha$ in the aforementioned choriocarcinoma cells, we realized that residual GCM1 activity is a key issue in establishing TS cells from term placentas. As a proof of concept, we successfully derived TS^{Term} cells from term placentas by suppression of GCM1 expression by hypoxia. We are concerned about the residual GCM1 activity in TS^{TC} and TS^{blast} cells as Cinkornpumin *et al.* had

to culture Okae's TS cells (hTSCs: TS^{CT1}, TS^{CT3}, and TS^{BLAST2} cells) under hypoxia in order to promote hTSC self-renewal but inhibit directed differentiation (*Stem Cell Rep* 15: 198-213, 2020). Whether TS^{CT} and TS^{blast} cells are true TS cell lines is an open question. We believe that the ground-state human TS cells should be GCM1-negative and are highly likely to constitute the pre-CTB population (GCM1 is barely expressed) in the single-cell RNA-seq study of 3D-cultured human blastocysts by Xiang *et al.* (*Nature* 577: 537-542, 2020). We appreciate the Reviewer's suggestion of using Okae's TS cell lines to study the GCM1- Δ Np63 α interaction. However, if these cells need to be maintained under hypoxia as Cinkornpumin *et al.* did, then the issue of few TS cells coexpressing GCM1 and Δ Np63 α remains. Regarding the loss of GCM1 function analyses, we have generated *GCM1*-knockout TS^{Term} cells and demonstrated that GCM1 is crucial for STB and EVT differentiation (Supplementary Fig. 5). Therefore, loss of GCM1 function in TS^{Term} cells results in loss of differentiation potency. Functional characterization of Δ Np63 α and GATA3 in the *GCM1*-knockout TS^{Term} cells is an intriguing topic; however, we think it is beyond the scope of the present study. *In situ* proximity ligation assay (PLA) is a method to identify physical closeness (<40 nm) of two proteins in cells. Because of its high sensitivity, we performed PLA to study the interaction between GCM1 and Δ Np63 α in TS^{Term} cells by careful calibration of the input normal IgG and primary antibodies. We cultured TS^{Term} cells under normoxic conditions for 96 h to maintain a population of Δ Np63 α - and GCM1-positive cells. Indeed, a significant increase in PLA signals were detected in the experimental group with GCM1 and Δ Np63 α antibodies compared with the control group with normal IgG and Δ Np63 α antibody (Fig. 4g). This observation supported that GCM1 very likely interacts with Δ Np63 α in TS^{Term} cells.

2. GATA3 is one of the earliest genes induced during trophoblast differentiation. The data and argument related to GATA3 regulation by deltaNP63 and GATA3-mediated negative regulation of GCM1 need further validations. The new transient transfection analyses with GATA3 minimal promoter is not a convincing experiment as normally the (-)500bp region enhancer is not an optimum regulator for GATA3 gene transcription. Thus, the data is a bit surprising. Authors should perform a chromatin immunoprecipitation analyses with Okae TS lines (where deltaNP63, GATA3 and GCM1 are all expressed) to show that deltaNP63 is binding to those putative binding motifs at the GATA3 locus. In the same TSC context, authors should show that loss of GATA3 is affecting GCM1 function (I am sorry, the earlier Scientific Reports paper includes experiments performed in HTR8 cell line, which does not represent trophoblast stem cell and the argument in the rebuttal letter about this aspect is

not a convincing one).

→ We thank the Reviewer for this constructive comment. Construction of the GATA3 promoter reporter vector in the present study was based on the study by Candi *et al.* (*J Cell Sci* 119: 4617-4622, 2006). In Fig. 4A of Candi's paper, $\Delta\text{Np63}\alpha$ was able to stimulate the GATA3 promoter activity. We have performed CHIP analysis to demonstrate that $\Delta\text{Np63}\alpha$ is associated with the GATA3 promoter in TS^{Term} cells (Supplementary Fig. 2c). We have also demonstrated that GATA3 expression is decreased by $\Delta\text{Np63}\alpha$ knockdown in TS^{Term} cells at the protein and mRNA levels (Supplementary Fig. 2d). Along this line, the expression of $\Delta\text{Np63}\alpha$ and GATA3 was decreased during the differentiation of TS^{Term} cells into STBs (Supplementary Fig. 2e). In addition, overexpression of exogenous GATA3-FLAG in TS^{Term} cells compromised STB differentiation (Supplementary Fig. 2f). Therefore, the $\Delta\text{Np63}\alpha$ -GATA3 axis is downregulated and GCM1 is upregulated in TS^{Term} cells during STB differentiation, which can be counteracted by GATA3 overexpression. As mentioned in our answer to comment#1, the antagonism between $\Delta\text{Np63}\alpha$ and GCM1 characterized in BeWo and JEG3 cells facilitated the derivation of TS^{Term} cells from term placentas. Indeed, key observations from the BeWo and JEG3 choriocarcinoma cells were also made in TS^{Term} cells in the present study. We think it is not practical to repeat all the BeWo and JEG3 experiments in TS^{Term} cells. Because GCM1 expression is suppressed in TS^{Term} cells under hypoxia, we anticipated that loss of $\Delta\text{Np63}\alpha$ or GATA3 will not further affect GCM1 expression in TS^{Term} cells. Just as a note of explanation, we used BeWo and JEG3 cells, but not HTR8 cells, to demonstrate suppression of GCM1 activity by GATA3 in our *Sci Rep* paper. We have tested HTR8 cells from different sources and found that HTR8 cells are unlikely to be trophoblast cells as claimed. Therefore, we are concerned about the validity of the published HTR8 data.

Additional Comments:

1. Introduction, line 66: References regarding Hippo signaling are selective and does not include manuscripts related to human TSCs. Authors should include following references.

Home P *et al.*, *Proc Natl Acad Sci U S A.* 2012 109(19):7362-7

Meinhardt G *et al.*, *Proc Natl Acad Sci U S A.* 2020 117(24):13562-13570

Saha B *et al.*, *Proc Natl Acad Sci U S A.* 2020 117(30):17864-17875.

→ We have incorporated the information of these references into the revised manuscript.

2. Introduction line 69-74, this is also a partial and incomplete information.

There are several recent manuscripts, which showed that the Naïve human ES cells can be reliably converted to the trophoblast stem cell fate using the Okae EGF/CASVY medium. Authors should include this aspect and associated references in the manuscript.

→ We have incorporated additional information and references about TS cell studies using the Okae EGF/CASVY medium in the revised manuscript.

Reviewers' Comments:

Reviewer #1:

Remarks to the Author:

The authors have not fully addressed my points #4 and #5 from the previous review. In addition, reading through reviewer #2's concern, I think a compromise may be reachable if the authors addressed the below major points:

1) I do not see an updated Figure 6a (based on the response to reviewer #1, point #4, it is missing Okae's TSC and their derivatives, along with first trimester CTB, STB and EVT). I do see the Okae TSC and their derivatives in Figure 6C and 6D, but not the first trimester CTB, STB and EVT? Was a wrong version of Figure 6a uploaded?

2) I don't see where the differences between TS-CT, TS-blast, and TS-term are discussed (in response to reviewer #1, point #5)? There is discussion of TMEM131L and RIPOR2 added in, but this doesn't address my original question of differences between the stem cell states of TSC derived from blastocyst, first trimester CTB, and term CTB.

3) I recognize that the authors have done much work here and applaud them for it; however, I also understand reviewer #2's concern regarding the use of choriocarcinoma cell lines (as well as non-trophoblast cell lines) for many of the questions. It would be worth mentioning this as a limitation in the discussion and note that future studies should try to make use of primary cells and TSC lines as much as possible.

Three minor points:

1) Figure 6a. the axes of this PCA are labelled MDS1, 2, and 3. I would suggest that the authors define MDS and provide the percent variation that each axis represents.

2) Figure 4h. Fold is misspelled "Flod"

3) Castel et al. cultured their iTSCs in hypoxia in preparation for EVT differentiation; so, I'm not sure this supports the claim that first trimester TSC or iTSC do better under hypoxia. Certainly all these cells were derived and cultured continuously under normoxia, so it is more likely that there are intrinsic differences between first trimester and term CTB which remain to be explored.

Reviewer #2:

Remarks to the Author:

The rebuttal letter and new experiments, which are incorporated based on my comments, are enough to satisfy my concerns about the manuscript. I have no more comments.

Response to the comments of Reviewer #1 and #2

REVIEWERS' COMMENTS

Reviewer #1 (Remarks to the Author):

The authors have not fully addressed my points #4 and #5 from the previous review. In addition, reading through reviewer #2's concern, I think a compromise may be reachable if the authors addressed the below major points:

1) I do not see an updated Figure 6a (based on the response to reviewer #1, point #4, it is missing Okae's TSC and their derivatives, along with first trimester CTB, STB and EVT). I do see the Okae TSC and their derivatives in Figure 6C and 6D, but not the first trimester CTB, STB and EVT? Was a wrong version of Figure 6a uploaded?

→ We truly appreciate Reviewer #1's comment. Our sincere apology to Reviewer #1 for uploading a wrong version of Figure 6a. The correct Figure 6a is now included in the revised Figure 6.

2) I don't see where the differences between TS-CT, TS-blast, and TS-term are discussed (in response to reviewer #1, point #5)? There is discussion of TMEM131L and RIPOR2 added in, but this doesn't address my original question of differences between the stem cell states of TSC derived from blastocyst, first trimester CTB, and term CTB.

→ We thank Reviewer #1 for this constructive comment. The gene expression profiles of TS^{Term} , TS^{CT} , and TS^{blast} cells are indeed very similar based upon PCA (Figure 6a) and Pearson correlation analysis (Figure 6c). Minor difference in the expression of lineage-specific genes was noted when the single-cell RNA-seq dataset of the 3D-cultured human blastocysts was incorporated into the heatmap analysis (Figure 6d). Nevertheless, we believe that the stem cell states of TS^{Term} , TS^{CT} , and TS^{blast} cells are also very similar because the differentiation potential of the three TS cell types is comparable.

3) I recognize that the authors have done much work here and applaud them for it; however, I also understand reviewer #2's concern regarding the use of choriocarcinoma cell lines (as well as non-trophoblast cell lines) for many of the questions. It would be worth mentioning this as a limitation in the discussion and note that future studies should try to make use of primary cells

and TSC lines as much as possible.

→ We thank Reviewer #1 for this comment. We have incorporated “Finally, it is worth mentioning that choriocarcinoma and non-trophoblast cell lines are used for characterizing the antagonism between $\Delta Np63\alpha$ and GCM1 in the present study. Future studies should try to make use of primary trophoblast cells and TS cell lines as much as possible to reinforce the conclusion of the present study.” into the last paragraph of revised Discussion.

Three minor points:

1) Figure 6a. the axes of this PCA are labelled MDS1, 2, and 3. I would suggest that the authors define MDS and provide the percent variation that each axis represents.

→ A correct PCA figure has been included in the revised Figure 6 as described in the major comment #1.

2) Figure 4h. Fold is misspelled "Flod"

→ We thank Reviewer #1 for this comment. We have corrected the typographical error in the revised Figure 4a.

3) Castel et al. cultured their iTSCs in hypoxia in preparation for EVT differentiation; so, I'm not sure this supports the claim that first trimester TSC or iTSC do better under hypoxia. Certainly all these cells were derived and cultured continuously under normoxia, so it is more likely that there are intrinsic differences between first trimester and term CTB which remain to be explored.

→ We thank the Reviewer #1 for this comment. According to the PCA data in the revised Figure 6a, there are differences in gene expression profiles for first-trimester and term CTBs. Interestingly, such differences seem to disappear when the CTBs were induced into TS cells. Indeed, the gene expression profiles for the TS cells derived from first-trimester and term CTBs become more similar to each other (Fig. 6a).

Comments from Reviewer #1 after seeing the previously requested data

The authors have addressed by first main point but not the second one.

Figure 6d (nor Figure 6a) address differences between their TS-term and the Okae TS-CT and TS-blast. To compare these directly, the respective STB and EVT derivates have to be removed and the TS lines compared directly

→ We thanks Reviewer #1 for this comment. According to the comments of Reviewer #1 and the Editorial team on our revised manuscript, the RNA-seq datasets of Okae's TS cells (and EVT and STB derivatives) as well as their primary CTBs, EVTs, and STBs together with the datasets of term CTB-derived TS cells (and EVT and STB derivatives) were included in transcriptome-wide comparisons using a PCA (Fig. 6a). In the Reviewer #2's major comment #10(vi) on our original manuscript, he or she requested a detailed comparison of actual gene expression levels in stem vs. differentiated states of the TS cells derived from term placenta with respect to the TS cells derived from first-trimester placenta. To this end, TS-, STB-, and EVT-specific genes were selected based upon Figure 6d for the gene expression comparisons requested by Reviewer #2 (Figure 6e). Therefore, we think the connection between Figures 6d and 6e will be obscure by removing the STB and EVT data in Figure 6d. We have rearranged the heatmap data in the revised Figure 6d for better visual comparison of the gene expression patterns between TS^{Term}, TS^{CT}, and TS^{blast} cells.

Reviewer #2 (Remarks to the Author):

The rebuttal letter and new experiments, which are incorporated based on my comments, are enough to satisfy my concerns about the manuscript. I have no more comments.

→ We thank Reviewer #2 for his or her efforts on the improvement of our manuscript.